# SARS-CoV-2 infection of human lung epithelial cells induces TMPRSS-mediated acute fibrin deposition

Rachel Erickson [1,6], Chang Huang [1,6], Cameron Allen [1,6], Joanna Ireland[1], Gwynne Roth [1], Zhongcheng Zou[1], Jinghua Lu[1], Bernard A. P. Lafont [2], Nicole L. Garza[2], Beniah Brumbaugh[3], Ming Zhao[4], Motoshi Suzuki[4], Lisa Olano [4], Joseph Brzostowski [1], Elizabeth R. Fischer[3], Homer L. Twigg III[5], Reed F. Johnson[2] & Peter D. Sun [1] ✉

Severe COVID-associated lung injury is a major confounding factor of hospitalizations and death with no effective treatments. Here, we describe a non-classical fibrin clotting mechanism mediated by SARS-CoV-2 infected primary lung but not other susceptible epithelial cells. This infection-induced fibrin formation is observed in all variants of SARS-CoV-2 infections, and requires thrombin but is independent of tissue factor and other classical plasma coagulation factors. While prothrombin and fibrinogen levels are elevated in acute COVID BALF samples, fibrin clotting occurs only with the presence of viral infected but not uninfected lung epithelial cells. We suggest a viral-induced coagulation mechanism, in which prothrombin is activated by infection-induced transmembrane serine proteases, such as *ST14* and *TMPRSS11D*, on NHBE cells. Our finding reveals the inefficiency of current plasma targeted anticoagulation therapy and suggests the need to develop a viral-induced ARDS animal model for treating respiratory airways with thrombin inhibitors.

COVID-19 is an acute respiratory disease caused by coronavirus SARS-CoV-2. Early autopsy of COVID-19 patients revealed the presence of extensive diffuse alveolar damage (DAD) and fibrin microthrombi in infected lungs[1–5], a likely cause of reduced oxygen intake and the need for ventilation in many hospitalized patients. Thrombotic structures were observed in pulmonary artery, vein, and microvasculature sites[6,7]. However, despite the use of low molecular weight heparin as a standard thromboprophylaxis to mitigate coagulopathy, mortality persists[7–9]. While the development of pulmonary DAD is a general feature of acute respiratory distress syndrome (ARDS) and is also observed in other respiratory infections, including fatal H1N1 influenza, SARS-CoV, and MERS infections[10–12], the mechanism of SARS-CoV-2 induced DAD is not well understood. Understanding of the mechanism of SARS-CoV-2 infection-induced alveolar damage would facilitate the development of therapeutic approaches to mitigate COVID-caused pulmonary disease and thus reduce the mortality rate associated with the pandemic. COVID-associated DAD exhibits characteristic hyaline membranes and fibrinous deposition in infected alveolar space[5,13–15]. These fibrin-containing membranes are thought to be the result of dysfunctional thrombosis through the classical coagulation pathway or inflammation-induced TGF-β driven myofibroblast activation[16,17], although other mechanisms have also been

[1]Laboratory of Immunogenetics, National Institute of Allergy and Infectious Diseases, National Institutes of Health, 5625 Fishers Ln, Rockville, MD 20852, USA. [2]SARS-CoV-2 Virology Core, Laboratory of Viral Diseases, National Institute of Allergy and Infectious Diseases, National Institutes of Health, Bethesda, MD 20892, USA. [3]Research Technologies Branch, National Institute of Allergy and Infectious Diseases, National Institutes of Health, 903 South 4th Street, Hamilton, MT 59840, USA. [4]Research Technologies Branch, National Institute of Allergy and Infectious Diseases, National Institutes of Health, 5625 Fishers Ln, Rockville, MD 20852, USA. [5]Division of Pulmonary, Critical Care, Sleep, and Occupational Medicine, Indiana University Medical Center, 1120 West Michigan Street, CL 260A, Indianapolis, IN 46202, USA. [6]These authors contributed equally: Rachel Erickson, Chang Huang, Cameron Allen. ✉e-mail: psun@nih.gov

proposed, such as the involvement of neutrophil extracellular traps[18]. The classical coagulation pathway involves sequential activation of a cascade of serine proteases leading to thrombin activation and the cleavage of fibrinogen to produce fibrin polymers[19,20].

Evidence shows that SARS-CoV-2 infections result in activation of host inflammatory antiviral response, including the type I interferon pathway and the expression of many inflammatory cytokines, as well as the infiltration of neutrophils and macrophages[21]. It is assumed that the overt host inflammatory response is responsible in part for COVID-associated lung injuries[22]. It is not clear, however, if the viral infection directly contributes to fibrinous depositions associated with DAD. Using primary human bronchial epithelial cells as a model system, we show that both SARS-CoV-2 pseudovirus and field strains induce fibrin clot formation from infected but not uninfected normal human bronchial/tracheal epithelial (NHBE) cells. The infection-induced fibrin clot formation is specific to the primary lung cells as the infection of either ACE2-expressing Vero or HEK293T cells failed to induce fibrin clot formation. Distinct from the classical plasma coagulation pathway, this infection-induced fibrin clot formation was cell-mediated, occurring inside of the alveolar space but outside of blood vessels and independent of many classical coagulation factors. This cell-mediated fibrin deposition required lung epithelial cell-expressed transmembrane serine proteases to activate prothrombin, resulting in fibrin clotting that can be inhibited by the existing pharmaceutical thrombin inhibitors, argatroban, and dabigatran. Importantly, infected but not uninfected NHBE cells produced fibrin clots in the presence of bronchoalveolar lavage fluid (BALF) derived from acute COVID patients but not BALF from healthy individuals.

Infection-induced fibrin deposition in alveolar space may undermine the effectiveness of heparin treatment used to target coagulopathy in blood vessels as standard hospital care of COVID-19 diseases. Identifying the mechanism by which SARS-CoV-2 induces acute fibrin deposition is important in developing new therapeutic treatments to reduce mortality associated with COVID-19 pandemic. Our finding of the infection-induced lung cell-mediated airway fibrin clotting suggests that targeting of infected lungs directly with thrombin inhibitors can be an effective therapeutic approach to mitigate SARS-CoV-2 lung pathogenesis.

## Results

### Elevated prothrombin and fibrinogen levels in COVID-19 lung fluid

COVID-19-associated lung injury was thought to be due in part to a dysregulated coagulation leading to thrombosis in veins as evidenced from frequent microthrombi formation in diseased lungs[3,7,23]. This is also supported by elevated plasma D-dimer levels in some patients with severe COVID-19[24–27]. However, despite the use of anti-coagulants such as heparin in hospitalized COVID patients, the clinical onset of COVID-associated lung damage continued to drive the mortality. In addition to microvascular thrombosis, hyaline membrane formation, a hallmark of acute respiratory distress syndrome (ARDS), was also frequently observed in hospitalized COVID patients, suggesting the presence of inflammatory exudate containing plasma-borne coagulation factors in infected alveolar space. Indeed, activated monocytes and macrophages as well as proinflammatory cytokines were detected in cells from bronchoalveolar lavage (BAL)[28]. However, the link between SARS-CoV-2 infection and fibrin deposition in infected lungs remains unclear[29].

To address SARS-CoV-2 infection-induced changes in protein content in COVID-19 lungs, we performed mass spectrometry-based proteomics analysis on bronchoalveolar lavage fluid (BALF) from three healthy, four acute (COVID-acute) and four recovered (COVID-recovered) individuals. The acute and recovered COVID samples were taken within a week of symptom onset and more than 30 days after discharge from hospital, respectively. Overall, the mass spectrometry proteomic

analyses identified between 400 and 900 proteins from each BALF sample with 55–80% overlaps (common proteins) between samples (Supplementary Fig. 1A, Supplementary Data 1). The overlaps in identified proteins correlated with their abundance, with the most abundant proteins showing greater than 90% overlap (Fig. 1A), suggesting rather similar compositions of enriched proteins in healthy, COVID-acute and COVID-recovered lungs. When the covariance in protein abundance was compared using a Pearson correlation coefficient analysis among a subset of 92 common proteins in all samples, it showed that protein abundances are more correlated among healthy as well as between the COVID samples but less correlated between healthy and COVID samples (Fig. 1B), suggesting SARS-CoV-2 infection resulted in systematic changes in protein enrichment in infected lungs. While both healthy and COVID BALF samples contained many enriched plasma proteins (Supplementary Data 1)[30], the abundance of proteins in several classes differed systematically between the samples. There is a clear increase of enriched plasma proteins in acute COVID samples compared to the healthy samples (Fig. 1C), suggesting an elevated infiltration of plasma into the infected lungs. The presence of inflammatory response in the COVID-acute sample is evident from the presence of C-reactive protein and increased abundance in complement components and major plasma proteins in the COVID-acute compared to the healthy samples (Supplementary Fig. 1B–D). Concurrently, the relative abundance of pulmonary surfactant-associated proteins and mucins appeared decreased in the COVID-acute samples compared to those of COVID-recovered and healthy samples (Supplementary Fig. 1C), consistent with a decreased pulmonary function in COVID-acute lungs. Several coagulation factors, including prothrombin, fibrinogen, FXII, FXIIIB, antithrombin III and plasminogen were identified by mass spectrometry (Fig. 1D, Supplementary Data 1)[29], and most of them were in higher abundance in the COVID-acute sample compared to the COVID-recovered and healthy samples.

To further quantify the inflammatory infiltration of fibrinogen and prothrombin during SARS-CoV-2 infection, we measured their concentrations together with total IgG from 15 healthy, 4 COVID-acute and 7 COVID-recovered samples using ELISA. Overall, the fibrinogen, prothrombin, and total IgG concentrations measured from COVID-recovered samples were not statistically different from those of healthy donors (Fig. 1E). In contrast, both fibrinogen and prothrombin were 50–100 fold elevated in the COVID-acute samples, consistent with their increased risk for fibrin deposition. Thus, compared to healthy lungs, the COVID-acute lung contained signatures of acute response protein, inflammatory infiltration of plasma proteins as well as innate immune components. The concentrations of many inflammatory proteins in the COVID-recovered samples were restored to levels similar to the healthy samples.

### Infected lung epithelial cells induced fibrin clot formation

Much of our understanding of COVID-associated lung injury is based on research of acute respiratory distress syndrome (ARDS)[31]. To investigate the link between viral infection and lung fibrin deposition, we used NHBE cells that are permissive to SARS-CoV-2 infection as a model system[32], and infected them with both replication incompetent SARS-CoV-2 pseudoviruses (pSARS-2), as well as replication competent field variants. All pSARS-2 viruses were generated by co-transfecting a variant-specific spike-expressing plasmid with a luciferase-expressing HIV core plasmid[33,34]. Indeed, both ACE2-expressing 293T and Vero cells as well as NHBE cells were readily infected by a SARS-CoV-2 pseudovirus (pSARS-2) expressing the prototypic Wuhan strain envelope spike protein (Fig. 2A, Supplementary Fig. 2A–C).

SARS-CoV-2 infections induce cellular and inflammatory responses in COVID lungs[35–38], but their relationship to fibrin deposition remains speculative. To this end, we adopted a turbidity-based fibrin clotting assay to measure the cleavage of fibrinogen and subsequent polymerization of fibrin[39,40] (Supplementary Fig. 2D). The 50–200-nm-

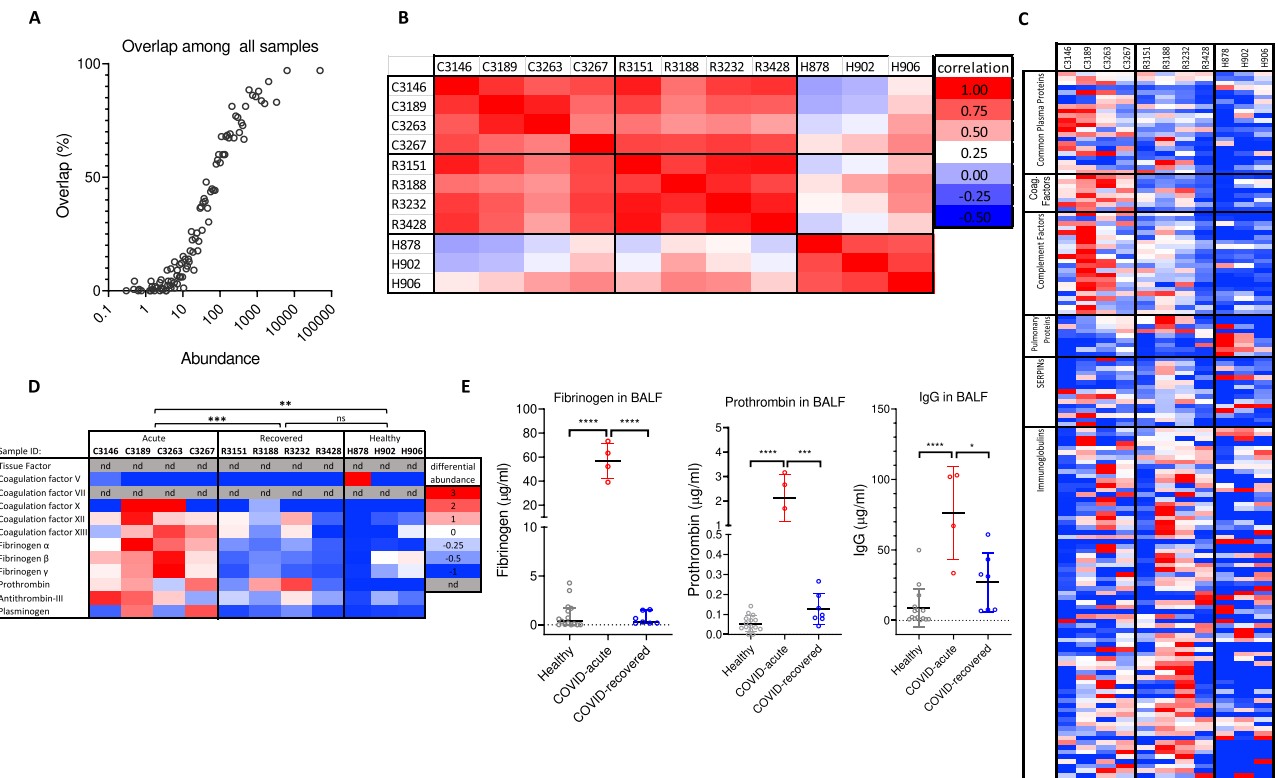

**Fig. 1 | Proteomics analyses of COVID BALF. A** The average overlap among all BALF samples decreases with identified protein abundance. Most abundant proteins exhibit greater than 80% overlap and are observed in all BALF samples whereas low abundance proteins show less overlap and more unique to each sample. **B** Pearson correlation coefficient between pairwise samples calculated based on the abundances of 92 common proteins among acute COVID (C), recovered COVID (R), and healthy (H) samples. **C** Heatmap showing differential protein abundance in mass spectrometry identified proteins among various BALF samples. Plasma proteins, complement components, and coagulation factors are upregulated in acute COVID BALF sample. **D**) Heatmap displaying the differential protein abundance for each coagulation factor. n.d. stands for not detected in the BALF sample. Panels (**C**) and (**D**) use the same color scheme. Statistical analyses were performed using two-way ANOVA with *p*-values significant for column analysis with $p = 0.0017$ (**), $p = 0.0002$ (***). **E** Concentrations of total IgG, fibrinogen, and prothrombin present in healthy, acute COVID and recovered COVID BALF samples as measured by ELISA. Data are presented as mean values +/−SD. Statistical analyses were performed using two-sided, unpaired t-tests, with $p = 0.0135$ (*), $p = 0.0003$ (***), $p < 0.0001$ (****).

thick fibrin fiber structures formed upon thrombin cleavage of fibrinogen are visible by confocal and electron microscopy[41–44] (Supplementary Fig. 2E, F). Fibrin clots formed by the extrinsic coagulation pathway in plasma are generally initiated with platelet aggregation and tissue factor activation[45]. To investigate if SARS-CoV-2 coronavirus infection could induce fibrin clot formation, we infected NHBE cells with the Wuhan pSARS-2 and then added fibrinogen to the infected cells (Fig. 2A). The result showed that the infected, but not uninfected NHBE cells induced fibrin clot formation (Fig. 2B). The fibrin clotting is most profound and dose dependent in pSARS-2 infections (Fig. 2C), diminished in VSV-pseudotyped virus infections and absent in envelope-null virus infections (Supplementary Fig. 3A, B). To further address if SARS-CoV-2 spike alone or the RBD binding to ACE2 on NHBE cells are sufficient to induce fibrin clot formation, we treated NHBE cells with soluble Wuhan spike trimer, S1 subunit or the ACE2 binding RBD domain and performed fibrin clotting assay after the treatment. While the omicron pSARS-2 infected NHBE cells induced fibrin clot formation, none of the soluble protein treated NHBE cells supported fibrin clotting (Supplementary Fig. 3C), suggesting that spike binding to NHBE cells is insufficient and viral infections are required for NHBE cells to induce fibrin clotting. The fibrin clot formation by infected NHBE cells may not be unique to SARS-CoV-2, however, as VSV-pseudotyped virus infection of NHBE cells also induced fibrin clotting (Supplementary Fig. 3A). The fibrin fibers formed in the presence of the infected NHBE cells were visible in both confocal and scanning electron microscopy images (Fig. 2D, E). Interestingly, many fibrin fibers were found to originate from NHBE cells in

the infected sample (Fig. 2F, Supplementary Fig. 3D), indicating a cell-mediated fibrin clotting mechanism induced by the viral infection. The pSARS-2 infection-induced fibrin clot formation, however, appears unique to lung epithelial cells as both infected NHBE and human small airway epithelial cells (HSAEC) induced fibrin clot formation (Fig. 2G). In contrast, neither infected Vero-E6 nor infected ACE2-293T cells induced fibrin clot formation (Fig. 2G, Supplementary Fig. 2A, B).

Further infections using alpha (UK), beta (South Africa), gamma (Brazil), delta and omicron variant spike-typed pSARS-2 viruses showed that this infection-induced fibrin clot formation was broadly observed in all pseudotyped variants (Fig. 3A). To address if fibrin clot formations can be induced by replication competent circulating strains of SARS-CoV-2 infections, we examined the infection of air−liquid interface cultured NHBE cells with the Washington (WA-1) strain of SARS-CoV-2 and found robust expansion of the virus in infected NHBE cells (Supplementary Fig. 4). Importantly, NHBE cells infected with circulating Washington (WA-1), alpha, beta and delta strains of SARS-CoV-2 supported fibrin clot formations in the infected but not uninfected NHBE cells (Fig. 3B). The infection-induced fibrin clots were visible by both confocal and scanning electron microscopy (Fig. 3C, D). The structures of these fibrin clots showed extensive fibrotic networks with dense fibers of 50–200 nm in thickness, similar to thrombin-induced fibers (Fig. 3C, D, Supplementary Fig. 2F). Thus, SARS-CoV-2 infections of primary human bronchial epithelial cells induced a cell-based fibrin clot formation, consistent with COVID-associated fibrin deposition in infected lungs.

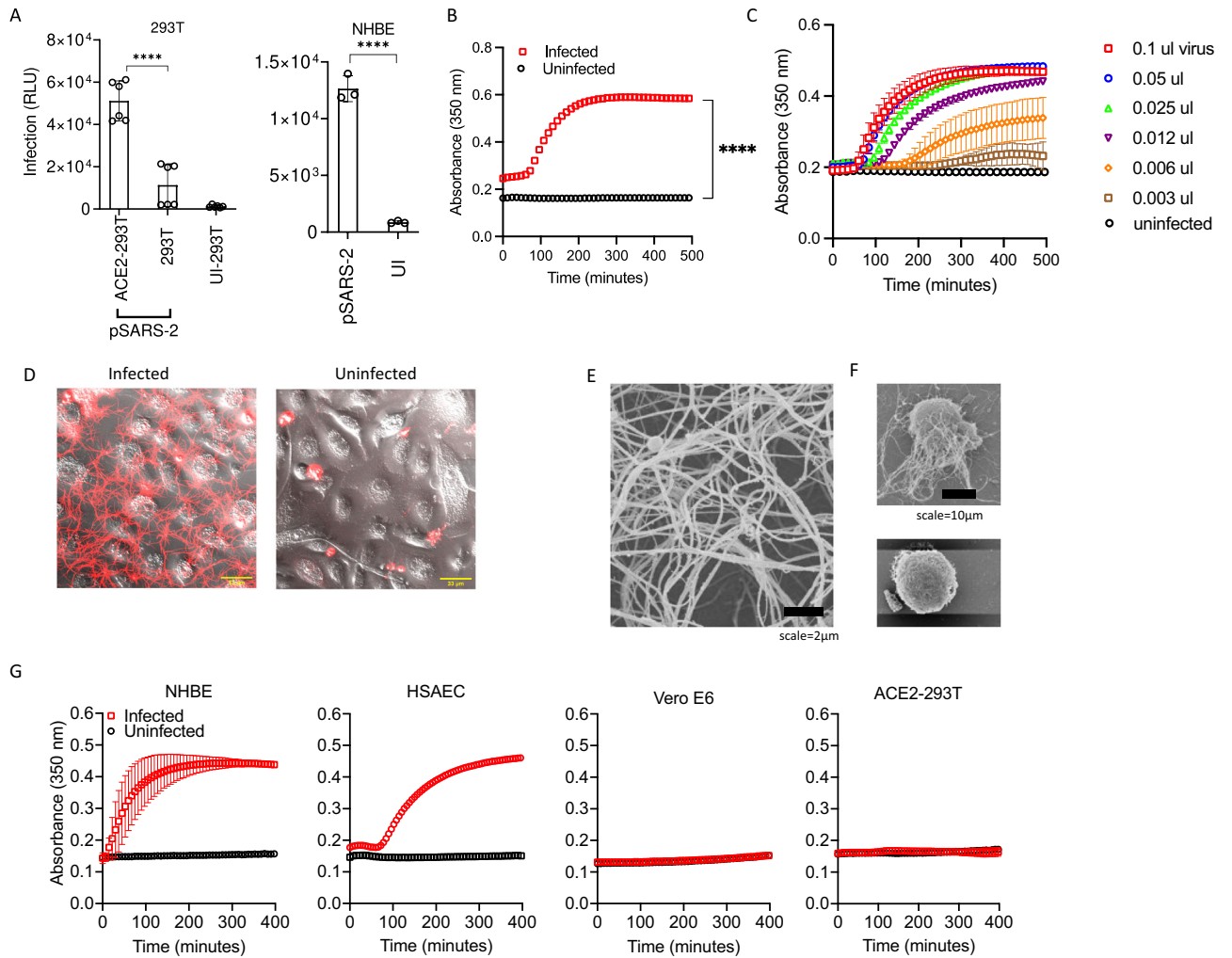

**Fig. 2 | SARS-CoV-2 pseudovirus infections and fibrin clot formations. A** SARS-CoV-2 pseudovirus infection of ACE2-293T, 293T, and NHBE cells. Cells were infected with 50 µl Wuhan strain of SARS-CoV-2 pseudovirus, approximately $5 \times 10^6$ copies of RNA/ml, for 48 h. Cells were lysed and infections were measured by luciferase activities. **B**, **C** Infected NHBE cells induced fibrin clot formation. NHBE cells were grown in 96-well (**B**) or 384-well plate (**C**) to near confluence and infected with 5 µl (**B**) or titration amount (**C**) of Wuhan pSARS-2 for 24 h before adding fibrinogen for clotting assay. **D** Confocal images of fibrin clot formation in NHBE infected with pSARS-2 or uninfected cells in the presence of fluorescently labeled fibrinogen. **E**) SEM images showing of fibrin network observed in infected NHBE sample. **F** Fibrin fibers associated with the infected (top) but not uninfected (bottom) NHBE cells. **G** Infected NHBE cells, but not Vero-E6 or ACE2-293T cells induce fibrin clot formation. All cells were infected with equal amount (4 µl each) of delta strain SARS-CoV-2 pseudovirus for 24 h before adding fibrinogen for clotting turbidity assay. Data shows means ± SD. All statistical analyses in this figure were performed using two-sided unpaired t-tests with *p*-value < 0.0001 (****).

It is not clear, however, if the fibrin clot formation induced by SARS-CoV-2 infection of NHBE cells required thrombin. To address this, we performed the fibrin clotting assays on pSARS-2 infected NHBE cells in the presence of a serine protease inhibitor, camostat, or a thrombin-specific inhibitor, hirudin. Both camostat and hirudin completely suppressed the infection-induced fibrin clot formation, similar to that of thrombin-induced clotting (Supplementary Fig. 5A, Fig. 4A–C). Similarly, two small molecular thrombin inhibitors, dabigatran and argatroban, also inhibited the infection-induced fibrin clotting (Fig. 4B, C). Consistently, hirudin also inhibited fibrin clotting induced by replication competent WA-1, beta and delta strains of SARS-CoV-2 infection of NHBE cells (Fig. 4D, E, Supplementary Fig. 5B), suggesting the infection-induced fibrin clotting is thrombin dependent. This thrombin-dependent fibrin clot formation by infected NHBE cells was a surprise as no thrombin was added to the infection and clotting assays. To address if thrombin was indeed involved in the infection-induced fibrin clotting, we performed mass spectrometry-based protein identification analyses on both infected and uninfected NHBE culture supernatants. Interestingly, multiple peptides derived from bovine thrombin were present in the infected NHBE culture supernatants (Fig. 4F), suggesting

a bovine additive in the cell culture media as a likely source of thrombin. To further address if the observed fibrin clotting by the infected NHBE cells was dependent on bovine prothrombin (bPT) and if human prothrombin (hPT) also supports the fibrin clotting, we depleted bovine prothrombin from the culture media using an anti-bovine prothrombin antibody and supplemented the media with 1 µg/ml human prothrombin. Depletion of bovine prothrombin from the culture media resulted in the loss of fibrin clot formation by the infected cells and subsequent supplement of human prothrombin to the bovine prothrombin depleted media restored the fibrin clotting in the infected but not uninfected cells (Supplementary Fig. 5C). Thus, the SARS-CoV-2 infection-induced fibrin clotting is thrombin dependent and occurs in the presence of human prothrombin.

## SARS-CoV-2 infection activates thrombin through TMPRSS family of proteases

Thrombin circulates as an inactive prothrombin in plasma, therefore it must be activated by coagulation factor Xa as part of the classical coagulation pathway[46]. It is not clear how prothrombin was activated during SARS-CoV-2 infection of NHBE cells. Interestingly, the culture

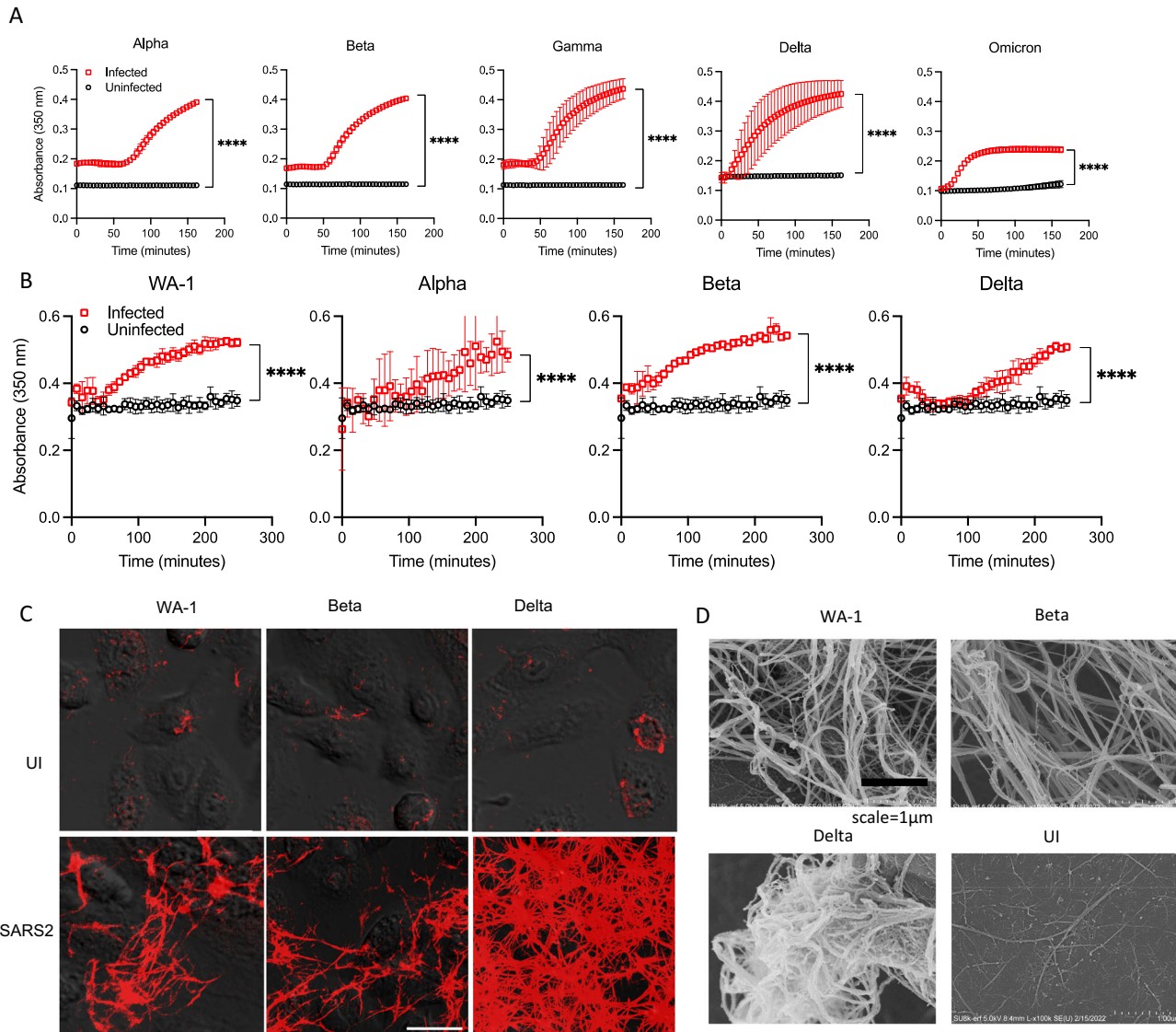

**Fig. 3 | Fibrin clot formation from NHBE cells infected with different variants of SARS-CoV-2. A** NHBE cells were grown in a 96-well plate and infected with 4 µl of different variant spike-typed pSARS-2 for 24 h before adding fibrinogen for clotting turbidity assay. **B** Fibrin clotting induced by replication competent SARS-CoV-2 variants. **C** Confocal images of fibrin clots observed in the presence of WA-1, beta and delta variants infected NHBE cells. **D** SEM images of fibrin clots in the presence of SARS-CoV-2 WA-1 or beta variants infected NHBE cells. Data shows means ± SD. Statistical analyses were performed using two-sided unpaired t-tests with p-values < 0.0001 (****).

supernatants from infected but not uninfected NHBE cells were sufficient to induce fibrin clot formation (Fig. 5A), suggesting that infected NHBE cells released proteases capable of functionally activating prothrombin. To address if the infected supernatant activated prothrombin, we synthesized a fluorogenic peptide corresponding to the factor Xa cleavage region of prothrombin, amino acids 324–333 (referred as Thrb-324). Factor Xa readily cleaved the prothrombin peptide, Thrb-324 (Fig. 5B). Importantly, the infected NHBE cell supernatant showed significantly higher cleavage of Thrb-324 than the uninfected supernatant (Fig. 5B). suggesting the ability of the infected supernatant to activate prothrombin. To further address if the infected supernatant contained activated thrombin, we measured enzymatic cleavage of a fluorogenic thrombin substrate, the N-terminal fibrinogen-β peptide corresponding to the thrombin cleavage region and is cleaved by thrombin but not prothrombin (Supplementary Fig. 5D). Consistently, the fibrinogen-β peptide was cleaved by SARS-CoV-2 infected but not uninfected NHBE supernatants and the cleavage was inhibited by thrombin inhibitor dabigatran (Supplementary Fig. 6A),

suggesting the presence of activated thrombin in the infected but not uninfected NHBE supernatants.

Although tissue factor (F3) was reported to be upregulated in SARS-CoV-2 infected NHBE cells[47], it is not clear if this resulted in the activation of classical coagulation pathway leading to the cleavage of prothrombin in our in vitro infection model. To investigate a possible involvement of tissue factor in SARS-CoV-2 infection-induced fibrin clotting, we performed transcriptome analyses using RNAseq on pSARS-2 infected NHBE and HSAEC cells as well as on a delta variant of SARS-CoV-2 infected NHBE cells. Both pSARS-2 and the field delta variant viruses upregulated type I interferon and innate antiviral responses in the infected NHBE and HSAEC cells compared to their uninfected controls (Fig. 5C). However, we did not observe significant upregulation in the tissue factor transcription in both pSARS-2 and the delta variant infected samples (Fig. 5C). Nevertheless, we performed tissue factor knockdown in both NHBE and HSAEC cells using a ribonucleoprotein complex-based CRISPR/Cas9 method (Fig. 5D)[48]. Despite a near complete elimination of the tissue factor expression,

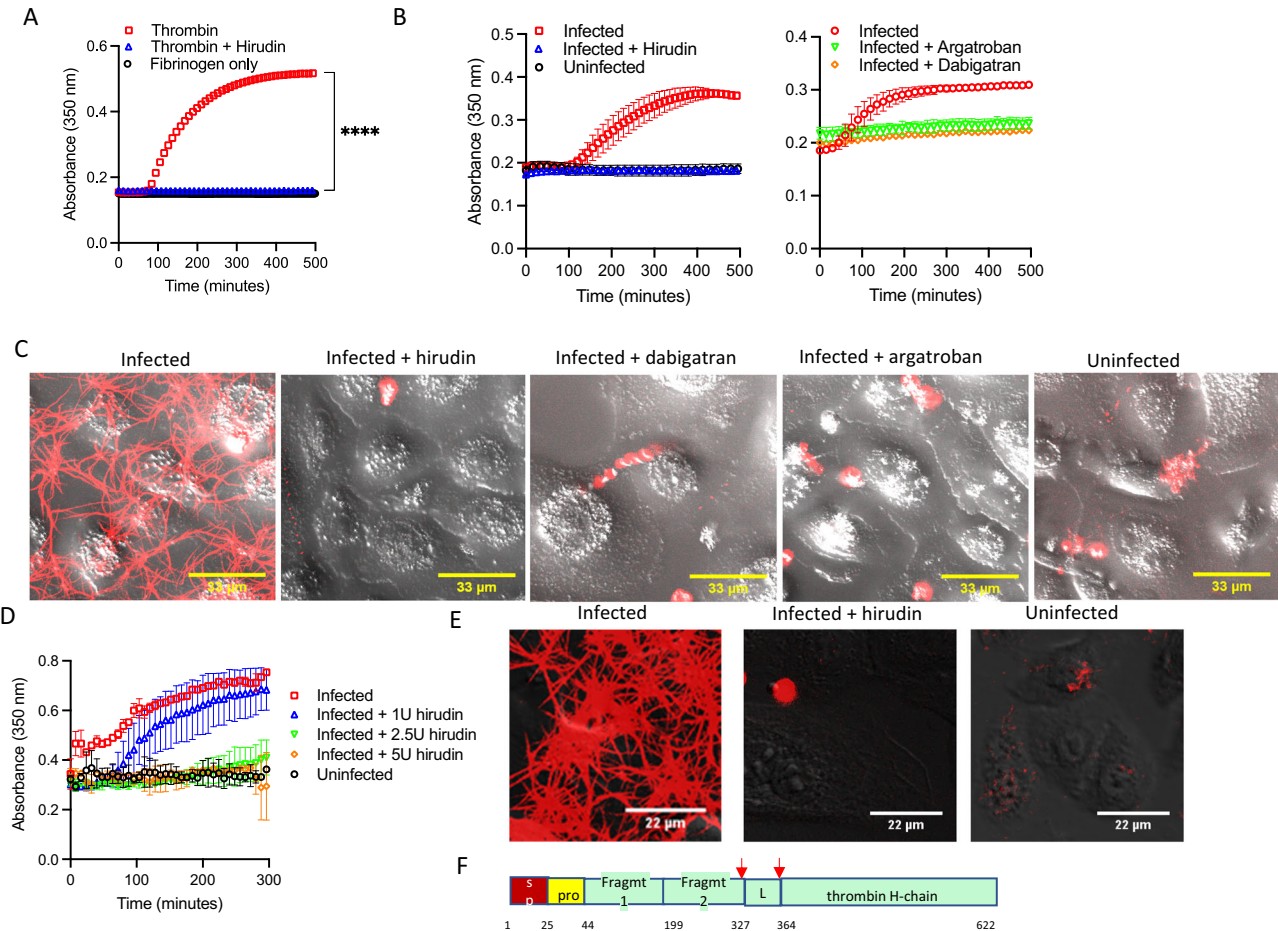

**Fig. 4 | SARS-CoV-2 induced fibrin clotting is thrombin dependent. A** Inhibition of Thrombin (-0.2 U/ml) induced fibrin clotting in the presence or absence of stoichiometric concentration of hirudin. The *p*-value was derived from two-sided, unpaired t tests with *p*-value < 0.0001 (****). **B** Wuhan SARS-CoV-2 pseudovirus infected or uninfected NHBE cells were assayed for fibrin clot formation in the presence of or absence of 5 U/ml hirudin, 5 μM dabigatran or 5 μM argatroban. NHBE cells were infected for 24 h with SARS-CoV-2. Hirudin were added to infected and uninfected cells during fibrin clotting assay. **C** Confocal images of fibrin clotting observed in pSARS-2 infected and uninfected NHBE cells in the presence of hirudin, dabigatran, and argatroban. Fluorescently labeled fibrinogen was added to cells 24 h post infection. **D** Fibrin clotting of WA-1 strain of SARS-CoV-2 infected NHBE cells in the presence of titrating amount of hirudin. **E** Confocal images of fibrin clotting observed in SARS-CoV-2 delta variant infected NHBE cells in the absence (left), presence of 5 U/ml hirudin (middle), and in uninfected cells. **F** Mass spectrometry identification of proteins in pSARS-2 infected NHBE cell culture supernatant. After 24 h infection with Wuhan variant pSARS-2 virus, NHBE cell culture media was removed and cells washed with PBS once before incubating them with PBS for 1 h to collect supernatants from both infected and uninfected NHBE cells for mass spectrometry analyses. Seven peptides were mapped to regions of thrombin catalytic domain as indicated by short bars from infected but not uninfected samples. Data in panels (**A**), (**B**), and (**D**) show means ± SD.

both knockdown NHBE and HSAEC cells formed fibrin clots in the presence of pSARS-2 infection (Fig. 5E), demonstrating that the infection-induced fibrin clot formation is independent of tissue factor-mediated coagulation.

As many fibrin fibers originated from infected cell surface (Fig. 2F, Supplementary Fig. 3D), we investigated the potential involvement of TMPRSS family serine proteases in prothrombin activation. NHBE and HSAEC cells express *ST14* and *TMPRSS11D* (Fig. 6A, B), and both are mildly upregulated in the presence of the viral infections (Fig. 5C). Interestingly, these TMPRSS genes are expressed in NHBE and HSAEC cells that support fibrin clotting but absent in Vero E6 and 293T cells that do not support fibrin clot formation (Fig. 6A, B, Fig. 2G). The protease domains of *ST14* and *TMPRSS11D* are referred to as matriptase and human airway trypsin-like protease (HAT)[49,50], respectively. *ST14* and *TMPRSS11D* are known to be upregulated in idiopathic pulmonary fibrosis[51,52]. The mouse homolog of *ST14*, epithin, was shown to be shed by ADAM17 in response to inflammatory stimulation and human matriptase activation required proteolytic cleavage[53–56]. Consistently, SARS-CoV-2 infection of NHBE cells released matriptase into cell culture supernatant (Fig. 6C), suggesting the infection activated the

protease. To address if the serine protease released by infected NHBE cells is the result of cell surface shedding, various protease inhibitors were added to NHBE cells during pSARS-2 infection. The infected supernatants collected in the presence of the protease inhibitors were assayed for fibrin clot formation. The results showed that the presence of metalloproteinase inhibitors BB-94 or prinomastat during the viral infection significantly reduced fibrin clot formation (Fig. 6D). To confirm that BB-94 did not inhibit the clotting step, we repeated the experiment but with protease inhibitors added post infection during the fibrin clotting. The result showed that BB-94 did not inhibit the fibrin clotting step (Supplementary Fig. 6B), suggesting the metalloproteinase inhibitors reduced the infection-induced cell surface shedding of TMPRSS proteins. To address if matriptase and HAT can activate prothrombin, we examined the cleavage of the fluorogenic prothrombin peptide Thrb-324 by recombinant catalytic matriptase and HAT. Both enzymes cleaved Thrb-324 similar to Factor Xa (Figs. 6E, 5B). Further, both enzymes promoted fibrin clot formation similar to Factor Xa in a component-based fibrin clotting assay by mixing the purified enzymes with human prothrombin and fibrinogen (Fig. 6F). To address if the expression of *ST14* or *TMPRSS11D* on cells is sufficient to

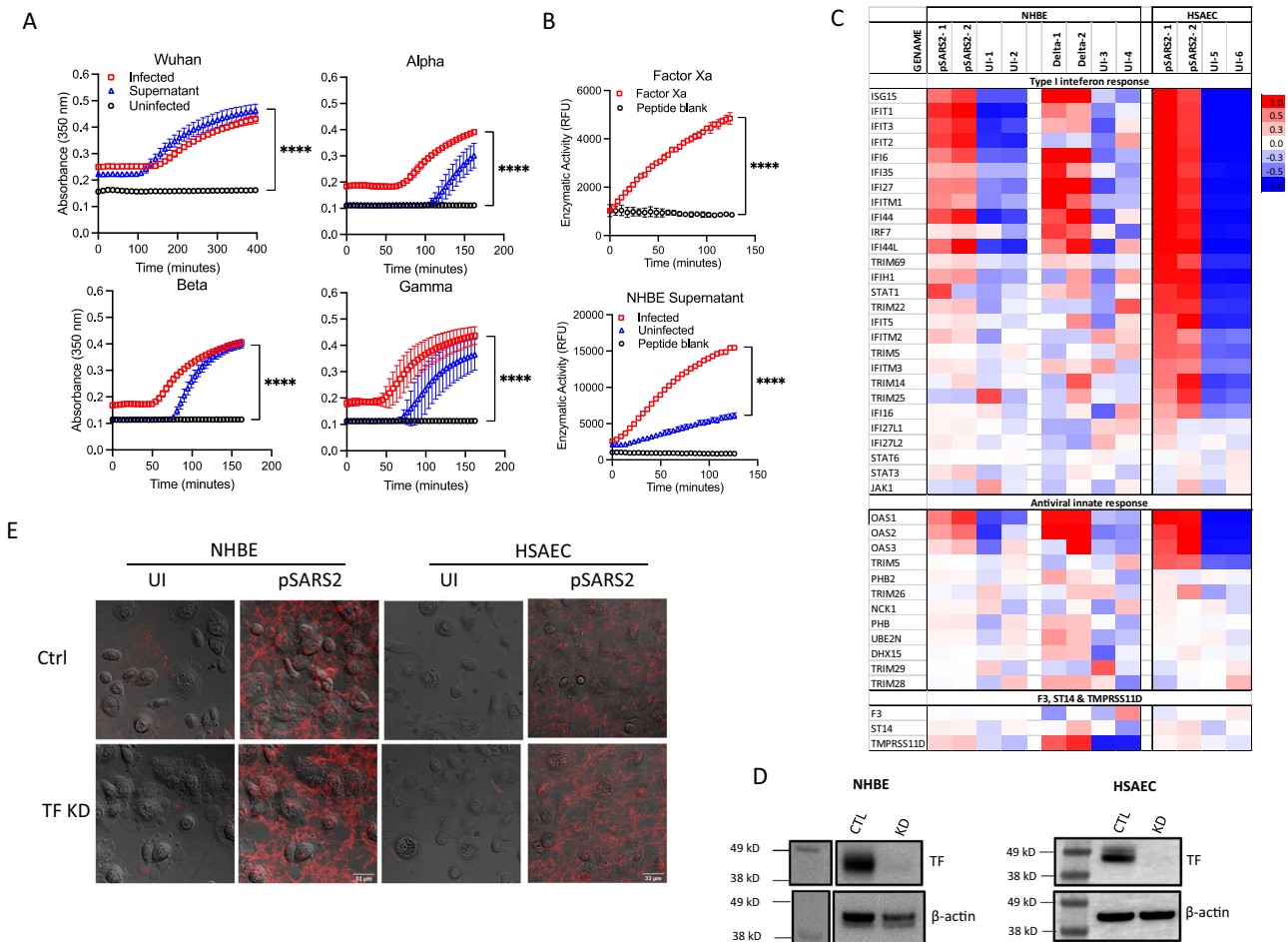

**Fig. 5 | Regulation of cellular pathway genes in SARS-CoV-2 infection. A)** Fibrin clot formation induced by NHBE cells and supernatant. NHBE cells were infected with $5 \times 10^6$ copies of RNA/ml of pSARS-2 variants for 24 h. Cell culture supernatants (50 μl) were transferred to separate wells. Fibrinogen were added to both supernatants and cells for clotting assay. **B** Enzymatic cleavage of fluorescent Thrombin-324 peptide by Factor Xa and NHBE supernatant. Factor Xa recombinant protein or supernatant from infected/uninfected NHBE cells were added to thrombin-324 peptide and enzymatic activity was measured by increase in fluorescence over time. Data in panels (**A**) and (**B**) show means ± SD with the *p*-values < 0.0001 (****) derived from two-sided, unpaired t-tests. **C** Differential gene expressions of various interferon, antiviral response genes as well as tissue factor (F3) and TMPRSS genes by RNAseq analyses using Illumina sequencing. NHBE or HSAEC cells were infected with either Wuhan spike containing pseudovirus (pSARS-1 and pSARS-2 for samples 1 and 2), or delta varion of field strain SARS-CoV-2 (samples delta-1 and delta-2) with their corresponding uninfected controls (UI-1 through UI-6). **D** Western blot showing the knock down of tissue factor in NHBE and HSAEC cells. **E** Fibrin clot formation induced by SARS-CoV-2 pseudovirus infection of tissue factor or control knockdown NHBE and HSAEC cells.

trigger infection-induced fibrin clotting, we transfected non-clotting ACE2-293T cells with plasmids encoding full-length *ST14* or *TMPRSS11D* genes, then infected the transfected cells with a delta variant of pSARS-2 for fibrin clotting assays. The results showed that the infection of either *ST14* or *TMPRSS11D* transfected but not untransfected ACE2-293T cells generated fibrin clots (Fig. 6G). Together, these results show that SARS-CoV-2 infection of NHBE cells induced shedding of TMPRSS proteins, such as matriptase and HAT, that are capable of activating prothrombin for fibrin clot formations.

**Infected NHBE cells induced acute COVID BALF to form fibrin clots ex vivo**

Our current work showed that SARS-CoV-2 infection of lung epithelial cells resulted in activation of membrane bound serine proteases, such as matriptase and HAT, that are capable of activating prothrombin and inducing fibrin clot formation. As the concentrations of prothrombin and fibrinogen are elevated in COVID-acute BALF (Fig. 1), the risk of fibrin deposition is higher in the acute samples than in the recovered or healthy BALF samples. It is not clear, however, if SARS-CoV-2 infection can trigger fibrin clot formation in BALF and whether the elevated levels of prothrombin and fibrinogen are sufficient to form fibrin clots

without the viral infection. To address this, we concentrated both healthy and COVID BALF samples 20 fold to approximate that of lung epithelial lining fluid[57], and performed the fibrin clotting assays in the presence of pSARS-2 infected or uninfected NHBE cells.

As expected, fibrin clots were readily detected in infected but not uninfected NHBE cells in the presence of added fibrinogen (Fig. 7A, top left). When replacing fibrinogen with BALF from healthy donors, no significant fibrin clot formations were detected regardless of the viral infections (Fig. 7A). However, three of the infected NHBE cells induced fibrin clots when exogenous fibrinogen was supplemented into the healthy BALF samples (H877, H880, H882), suggesting the fibrinogen concentrations in the healthy BALF are insufficient to induce fibrin clotting. In contrast to the healthy BALF, three of the COVID-acute BALF (C3263, C3267, and C3189) supported infected NHBE cells to form fibrin clots without addition of fibrinogen (Fig. 7B). Interestingly, visible fibrin clots were observed in uninfected NHBE cells in the presence of C3263 BALF, suggesting the presence of prior activated thrombin in this BALF sample. In all three acute COVID cases, the viral infection either triggered or enhanced fibrin clot formations, illustrating the viral contribution to BALF fibrin clot formation. Consistent with their lower concentrations of fibrinogen in recovered COVID

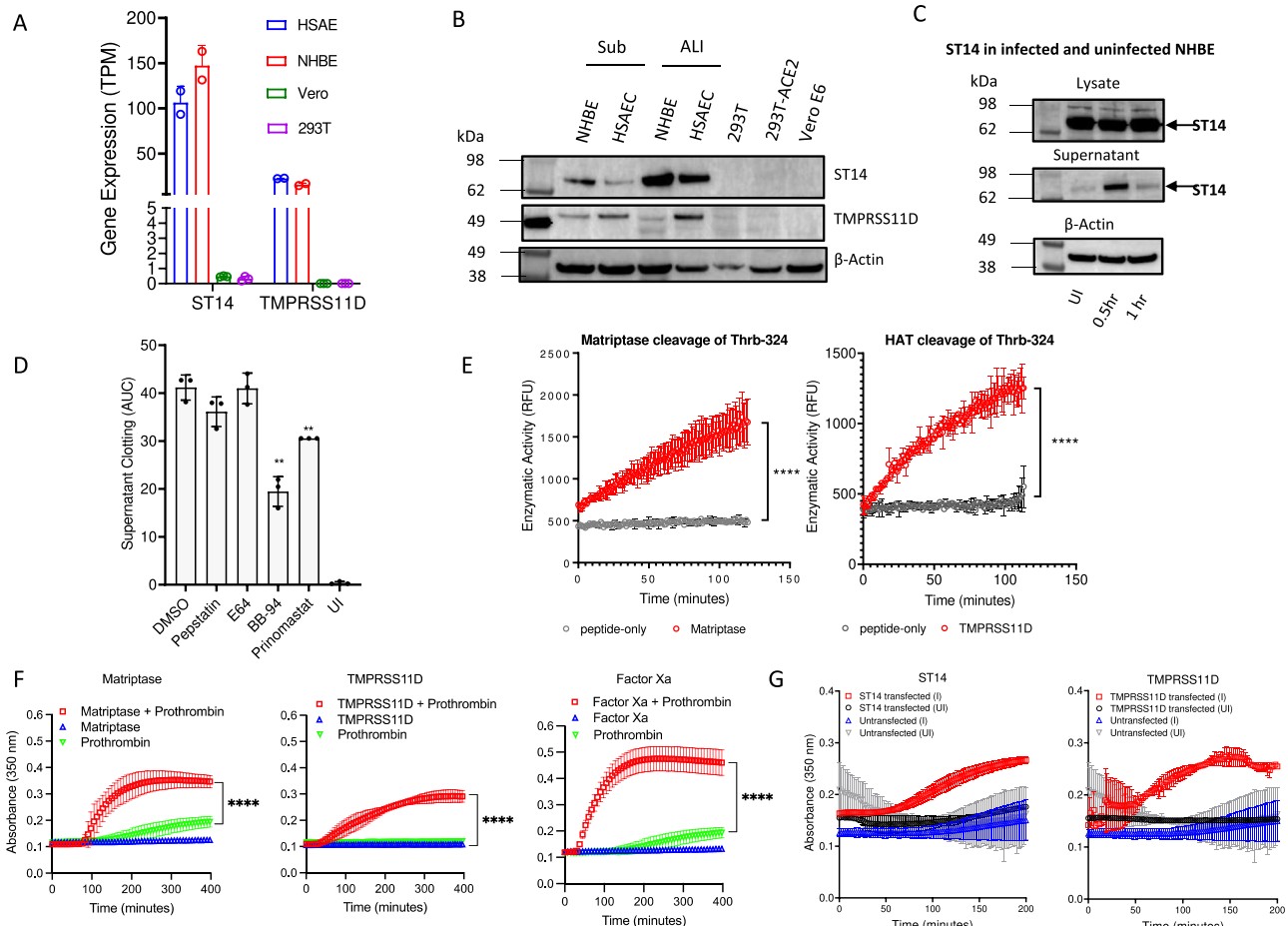

**Fig. 6 | Contribution of matriptase and HAT to fibrin clot formation.**
**A**, **B** Expression of members of TMPRSS genes, ST14 and TMPRSS11D, in various cells as measured by **A** counts per 10 million total reads (TPM) from RNAseq, and **B** western blot. The expressions of TMPRSS genes in HEK293 and Vero cells are derived from GSE153744 and GSE83900 datasets from NCBI GEO databases. **C** pSARS-2 infection resulted in the release of soluble ST14 in the culture supernatant. **D** Metalloproteinase inhibitors, BB-94 and prinomastat, but not others inhibited infected NHBE cells induced fibrin clotting. The inhibitors were added

during the viral infection but not during fibrin clotting assay. **E** Enzymatic cleavage of the prothrombin peptide, Thrb-324, by recombinant matriptase and HAT. **F** Recombinant matriptase and HAT cleaved prothrombin for fibrin clot formation similar to factor Xa. **G** Infection of 25 ng ST14 or 50 ng TMPRSS11D transfected ACE2-293T cells induced fibrin clot formation. Infected (I) or uninfected (UI) ACE2-293T cells without transfection did not form fibrin clots. All data are presented as mean values ± SD and unless stated all statistical analyses are performed using two-sided unpaired student t-test with p-values < 0.05 (*), <0.01 (**), <0.0001 (****).

BALF samples (Fig. 1E, Supplementary Fig. 7), a majority of the COVID-recovered samples did not support fibrin clots with or without the viral infection (Fig. 7B, C). Unlike the healthy BALF tested, fibrin clot formation was visible in two COVID-recovered samples, R3232 and R3248, in the presence but not absence of the viral infections (Fig. 7C), suggesting a potential risk of fibrin clotting in some recovered COVID lungs. Interestingly, both R3232 and R3248 exhibited higher fibrinogen and prothrombin concentrations than other recovered COVID samples despite being significantly lower than those in acute COVID samples (Supplementary Fig. 7). These findings showed that SARS-CoV-2 infection of lung epithelial cells induced fibrin clot formations in most acutely infected lung fluids.

It is worth noting that not all acute COVID BALF showed equal fibrin clot formation. Despite the presence of elevated level of fibrinogen in C3146 BALF (Fig. 1E, Supplementary Fig. 7A), no significant fibrin clot was detected in infected NHBE cells (Fig. 7B). Similarly, despite supplementing with exogenous fibrinogen, no fibrin clots were observed in BALF from two of the healthy donors (H879 and H883) (Fig. 7A), suggesting fibrinogen may not be the only factor controlling fibrin formation, and there may be other fibrinolytic factors, such as plasminogen, present in BALF to suppress fibrin polymerization. Other anti-coagulation factors, such as antithrombin-III and serine protease inhibitors (SERPIN) were also found in both healthy and COVID BALF

(Supplementary Data 1, Supplementary Fig. 1E), and the levels of plasminogen and antithrombin-III were also increased in the acute BALF C3146 (Fig. 1D). Together, these data showed that SARS-CoV-2 infection of lung epithelial cells induced a cell-mediated fibrin clotting in alveolar fluid that potentially account for acute DAD observed in severe COVID lungs.

## Discussion

The classical coagulation pathway refers to a sequential activation of a network of serine proteases leading to thrombin-mediated fibrin clotting, a critical process to prevent excessive bleeding during wound healing. Dysregulated thrombosis, such as venous thromboembolism (VTE), is known to contribute to morbidity and mortality in cancer patients[58,59]. For COVID-associated lung pathology, various mechanisms, including TGF-β mediated extracellular collagen fiber formation and neutrophil extracellular traps[18,60], have been proposed. A widely accepted mechanism attributed fibrotic lung damage to inflammatory activation of the classical extrinsic coagulation pathway and its leakage through blood lining endothelial cells to infected lungs[16]. This is further exacerbated by increased tissue factor expression found in infected NHBE cells[47]. Here, our findings demonstrate a cell-mediated fibrin deposition that occurs in alveolar airway space independent of plasma coagulations (Supplementary Fig. 8A). While SARS-CoV-2

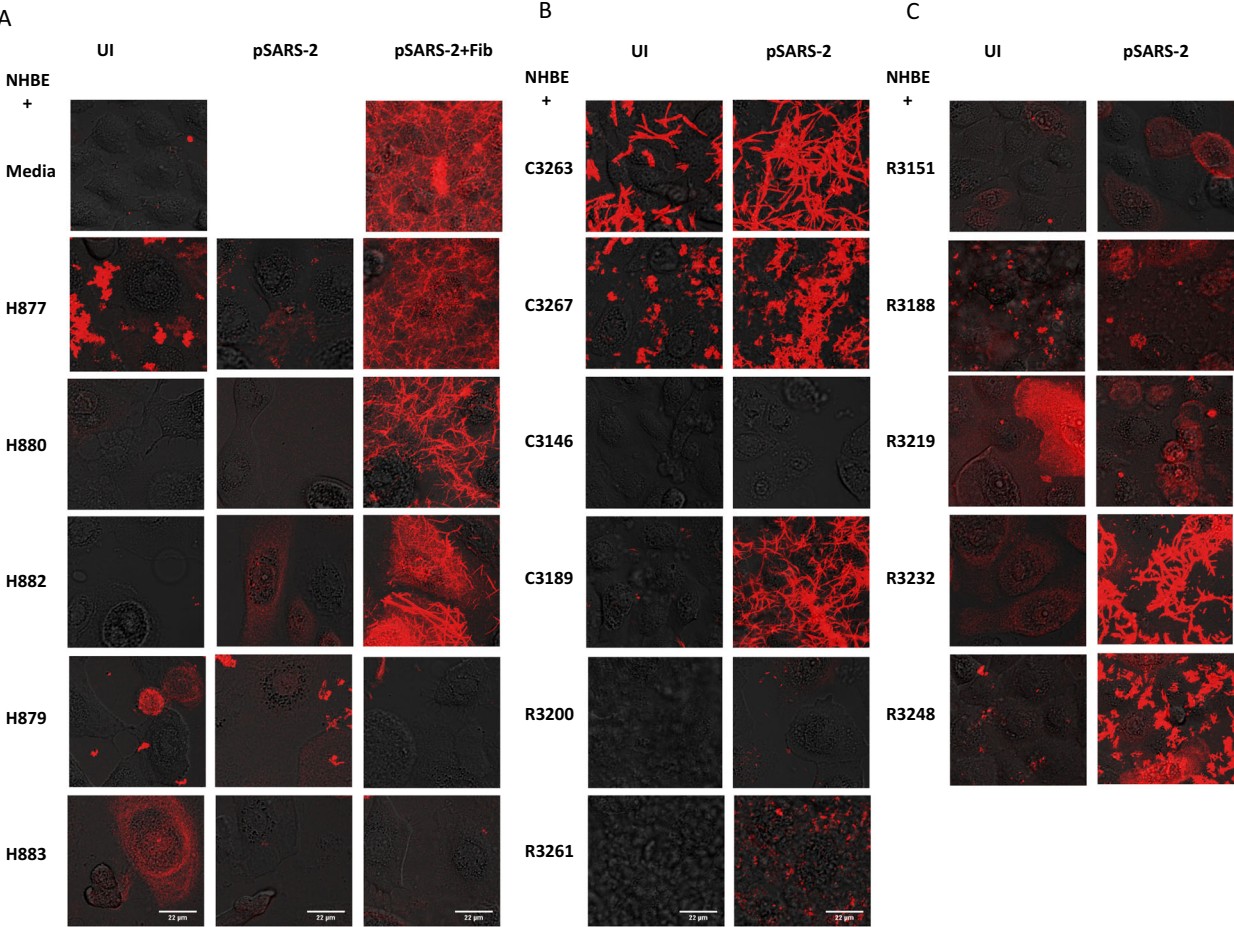

**Fig. 7 | SARS-CoV-2 infection promoted fibrin clotting in COVID BALF.** Delta variant pSARS-2 infected (pSARS-2) or uninfected (UI) NHBE cells were incubated with fibrinogen or various healthy (H877, H880, H882, H879, H883) (**A**), COVID-acute (C3263, C3267, C3146 and C3189) (**B**), and COVID-recovered (R3200, R3261, R3151, R3188, R3219, R3232 and R3248) (**B**, **C**) BALF samples in fibrin clotting assays. The formation of fibrin clots was observed using confocal microscope with incorporation of sub-stoichiometric amount of fluorescent TAMRA labeled fibrinogen.

infection resulted in increased infiltration of plasma proteins in infected alveolar space, we found that the increase in coagulation factors alone, including prothrombin and fibrinogen, was not sufficient to induce fibrin deposition. Instead, our results support an alternative activation of thrombin-mediated fibrin deposition that involves transmembrane serine proteases, such as *ST14* and *TMPRSS11D* from infected lung epithelial cells. In this model, SARS-CoV-2 infection of lung epithelial cells results in the activation of type II transmembrane serine proteases (TMPRSS) family members. The activated TMPRSS genes possess factor X-like activities to cleave prothrombin into activated thrombin, which in turn cleaves fibrinogen causing fibrin deposition in infected alveolar space (Supplementary Fig. 8B). While the infected cell culture supernatants exhibited enhanced cleavage of prothrombin peptide and the infected NHBE cells shed soluble *ST14* (Figs. 5B, 6C), trace amount of the extracellular fragment of *ST14* and *TMPRSS11D* could be found in the COVID BALF samples (Supplementary Data 1). Importantly, this infection-induced fibrin clotting occurs in the absence of tissue factor expression and does not require other serine proteases in plasma coagulation pathways.

The concentration of prothrombin and fibrinogen in BALF varied considerably between healthy and COVID individuals. The highest concentrations were found in acute COVID BALF at ~1–3 µg/ml and ~20–60 µg/ml for prothrombin and fibrinogen, respectively, suggesting their concentrations in lung epithelial lining fluid to be close to 20–60 µg/ml and 400–1200 µg/ml, respectively. These concentrations

of prothrombin and fibrinogen from acute COVID lungs are ~20–100x higher than those from healthy and COVID-recovered lungs, but they are in the same concentration range used in our experimental clotting assays. Consistently, the healthy and most COVID-recovered BALF did not form fibrin clots in the presence of infected NHBE cells. While fibrin clot formations were observed in 3 of 4 acute COVID BALF in the presence of SARS-CoV-2 infection, showing a significant risk of fibrin clots in acute COVID lung fluids, one of the acute COVID samples, C3146, failed to generate fibrin clotting in the presence of the viral infection, suggesting other factors may also influence the fibrin clot formation in the infected lung. This is also consistent with the fact that not every severe COVID patient succumbs to the disease. The direct contribution of the viral infection to fibrin clotting was evident as minimum or no clotting were detected in COVID-acute BALF in the absence of the viral infection. Interestingly, low level of fibrin clot formation was detected in C3263 in the absence of the viral infection, suggesting the presence of activated thrombin in this sample. Since none of the healthy nor recovered BALF showed fibrin clot formation, it suggests that the presence of activated thrombin in C3263 BALF is related to the SARS-CoV-2 infection in the patient. The clinical risk of developing pulmonary pathology associated with COVID has not been well characterized, although preexisting pulmonary conditions, severity of infection and the presence of inflammatory factors appear to predict the risk of COVID-associated lung damage[61–66]. Our findings suggest the concentrations of prothrombin, fibrinogen and other coagulation factors in SARS-CoV-2 infected lung fluids may be used to

indicate the risk of COVID-associated lung pathology. It is worth noting that while SARS-CoV-2 infected primary human lung epithelial cells induced fibrin clot formation in acute COVID lung fluid, it remains to be seen if this occurs in vivo. However, despite the presence of several SARS-CoV-2 susceptible animal models, including various ACE2-transgenic mice, Syrian hamster and rhesus macaque[67–69], they generally lack viral ARDS-like lung pathology observed in fatal human cases[70]. They are valuable for vaccine and antiviral development, but the infected animals rarely develop DAD that is characteristic of ARDS. Our study here is primarily based on an in vitro primary lung cell model with the results validated using ex vivo COVID-patient BALF. The lack of a suitable animal model hampers further in vivo validation of the cellular mechanism linking fibrin deposition, hyaline membrane formation and DAD.

The current use of heparin family of anticoagulants, while beneficial, failed to mitigate severe COVID-associated lung damage and mortality[7–9]. Heparin-related compounds target primarily activated clotting factor Xa with partial inhibition of thrombin activity[71]. Intravenous or subcutaneous injection of low molecular weight heparin has been used to prevent microvascular thrombosis in hospitalized COVID patients[8,9]. Our work revealed a SARS-CoV-2 infected NHBE cell-triggered fibrin clotting mechanism that occurs in the alveolar space outside of blood circulation and is independent of tissue factor in classical coagulations. Thus, administration of heparin targeting factor Xa intravenously may be less effective. Instead, we suggest potential more effective therapeutic interventions focusing on using nebulized direct thrombin inhibitors to target airway space. It would be interesting to see in a suitable animal model if nebulization in vivo indeed offers a better therapeutic outcome than intravenous injection in treating SARS-CoV-2 infected lungs.

## Methods

### Ethics statement
Bronchoalveolar lavage fluid (BALF) from healthy donors were purchased from Audubon Biosciences with informed consent (New Orleans, LA). BALF from COVID-experienced donors was collected with individual informed consent at Indiana University through a CLIA approved clinical BAL laboratory. The samples were obtained for clinical indications in patients with acute and post-COVID lung disease under an IRB approved protocol 1011003397R010 at Indiana University. This protocol allows storage of left-over clinical lavage samples for future research studies. All samples were deidentified before analyses.

### Cells and viruses
Normal Human Primary Bronchial/Tracheal Epithelial (NHBE, catalog PCS-300-010) and Small Airway Epithelial cells (HSAEC, catalog PCS-301-010), Vero E6 (catalog CRL-1586) and HEK 293T (catalog ACS-4500) cells were purchased from American Type Culture Collection (ATCC), Manassas, VA and cultured according to the manufacture's guidance. In particular, NHBE cells were cultured in Airway Epithelial Cell Basal Media (ATCC, PCS-300-030) supplemented with Bronchial/Tracheal Epithelial Cell Growth Kit (ATCC, PCS-300-040) under standard tissue culture conditions (37 °C and 5% $CO_2$). NHBE cells were harvested by washing with Dulbecco's phosphate-buffered saline (DPBS) (ATCC, 30-2200), then incubated with trypsin-EDTA (Life Technologies Corp, NY) at 37 °C for 5 min. The cells were resuspended in Airway Epithelial media for continued passage or cryopreservation. Cell counts were performed using a Guava Muse Cell Analyzer according to manufacturer's protocol (Luminex, TX). ACE2-expressing 293T cells (Catalog SL221) were purchased from Genecopoeia, Rockville MD. The culturing of NHBE cells in air−liquid interface was performed according to the manufacturer's instructions (Stemcell Technologies). Briefly, $3.3 \times 10^4$ or $4.5 \times 10^5$ cells in 0.2 or 3 mL PneumaCult™-Ex Plus Medium were plated in each transwell insert of 24- or

6-well plates (Corning, 3413 or 3450) with 0.5 or 3 mL of the same medium added into the basal chamber. After 2–3 days when confluence was reached, the medium from both the basal and apical chambers was removed and 0.5 or 3 mL of PneumaCult™-ALI Maintenance Medium (Stemcell Technologies, 05001) was added to the basal chamber and cells were cultured for 28 days with media change every 1–2 days. Beginning in week 2 post-airlift, mucus was removed from the apical surface by washing the cells with DPBS.

Circulating variants of SARS-CoV-2 viruses were expanded and characterized accordingly[72]. B.1.1.7 (alpha variant) and Washington-1 isolates were provided by BEI resources (Manassas, VA), B.1.351 (beta) and B.1.617.2 + AY.1 + AY.2 (delta) variants were kind gifts from Dr. Andrew Pekosz of Johns Hopkins University, Baltimore, MD.

For fibrin clotting assays, BALF samples were dialyzed against 0.045% of NaCl solution overnight to remove excess salts and then concentrated 20-fold using a speedvac (Labconco CentriVap) concentrator with the heating turned off.

### Production of SARS-CoV-2 pseudoviruses
For the production of the pseudovirus, HEK 293T cells were plated at a density of $2.5 \times 10^6$ per 10 cm plate and incubated at 37 degrees and 5% $CO_2$ overnight. Cells were co-transfected with a SARS-CoV-2 spike protein plasmid and an HIV NL4-3 env-nef-luciferase core using Lipofectamine 3000 according to the manufacturers protocol[33,73]. Plasmids encoding SARS-CoV-2 spike genes, including Wuhan[74], alpha (B.1.1.7)[75], beta (B.1.351), gamma (Brazil strain), delta (B.1.617.2), and omicron (B.1.1.529) BA.1 and BA.2 strains were obtained from Addgene. Supernatant containing pseudovirus particles was harvested 48 h post transfection and concentrated 100-fold using PEG-it Virus Precipitation Solution (System Biosciences, CA). The concentration of SARS-CoV-2 pseudovirus were estimated by RT-PCR in numbers of RNA copies/ml. In brief, RNA was extracted from 50 µl concentrated pseudovirus using the Qiagen RNeasy Mini Kit, and cDNA were generated using a C1000 Touch Thermal cycler (BIO-RAD, CA 94547) with ABI High-Capacity cDNA Reverse Transcription Kit following the manufacturer's protocol. HIV-1 NL4-3 LTR was amplified using TaqMan HIV-1 LTR primer/probe sets (Pa03453409_s1) from ThermoFisher with 50 ng cDNA as template. Samples were run in duplicate using a QuantStudio 6 Pro Real-Time PCR System (ThermoFisher, MA 02451) together with a serial dilution of a known copy number HIV DNA as standards. The pseudovirus concentrations were between $10^8$ and $10^9$ copies of RNA/ml. The infectivity of SARS-CoV-2 pseudovirus was examined by a luciferase assay, in which ACE2 expressing 293T cells or NHBE cells were plated in 96-well plates and grown to near confluence. The cells were infected with titration volume of the pseudoviruses between $5 \times 10^6$ and $5 \times 10^5$ copies of RNA/ml in their growth media. Polybrene was added at 5 µg/ml concentration for luciferase assay of SARS-CoV-2 pseudovirus infection of NHBE and Vero cells. Luciferase activity was assayed after 48 h of infection using Luc-Pair firefly luciferase HS assay kit according to the manufacture's protocol (Genecopoeia, Inc), and luminescence was measured by Synergy_H1 plate reader (BioTek, Inc). All field strains of replication competent SARS-CoV-2 viruses were expanded by infecting TMPRSS2-expressing Vero-E6 cells in NIAID BSL-3 facility. For infection with the field strains SARS-CoV-2, NHBE cells were infected with an MOI of 3 with isolates, WA1/2020 (Washington), B.1.351 (South Africa), or B.1.1.7 (UK) for 24 h prior to clotting assays. The use of pseudoviruses and replication competent strains in this study was approved by NIH inter-institute Biosafety committee (IBC) under protocol RD-20-VI-12.

### Infection of NHBE cells with SARS-CoV-2 pseudoviruses
For infection-induced fibrin clotting assay, NHBE cells growing at 60–80% confluence were infected with SARS-CoV-2 pseudovirus at doses between 0.05–4 µl virus per 10,000 cells, or between 1–40 copies of viral RNA per cell, for 24 h prior to clotting assays. For

infection with replication competent field strain SARS-CoV-2, the viruses were used at an MOI of 3. The infected supernatant was then removed and replaced with fibrinogen containing clotting buffer.

For transfection of TMPRSS genes, ACE2-expressing HEK 293T cells (Genecopoeia, Inc. MD) were plated at a density of 40,000 cells per well in a 96-well plate and incubated at 37 degrees, 5% CO₂ in DMEM growth media supplemented with 10% FBS for overnight. Plasmid encoding *ST14* (OHu19145C) or *TMPRSS11D* (OHu04628C) were synthesized in pcDNA3.1 vector with eGFP attached to N-terminus of the genes (GenScript). Cells were transfected with either *ST14* or *TMPRSS11D* plasmids using Lipofectamine 3000 according to the manufacturers protocol. Transfected cells were cultured with fresh media for 48 h and infected with titration amount of pseudovirus in cell culture media. After overnight infection, the cell culture supernatants were used in the fibrin clotting assay.

## Fibrin clotting turbidity assay

Purified fibrinogen from human plasma (Sigma-Aldrich, MO) was dissolved in 100 mM NaCl, 20 mM HEPES buffer. The solution was incubated at 37 °C for 10 min, then filtered through a 0.45 µm syringe filter. The solution was stored at 4 degrees for 30 min, then filtered again to remove aggregates. Concentration was measured using nanodrop, then the solution was aliquoted and frozen at −20 °C. Clot formation was assayed using fibrinogen solution diluted to 1.5 µM concentration in clotting buffer (20 mM HEPES, 137 mM NaCl, 5 mM CaCl₂). Diluted fibrinogen was added to thrombin enzyme (5 U/mL, Sigma) (positive control) or infected/uninfected NHBE cells seeded in a 96-well plate at 10,000 cells/well, or in a 384-well plate at 2500 cells/well for overnight. The absorbance was measured at 350 nm wavelength continuously with 2 min intervals for 4–10 h with Synergy_H1 (BioTek) plate reader. Fibrin clot formation causes scattering of light that passes through the solution, which increases the turbidity. To deplete bovine prothrombin in the cell culture media, 50 µg of sheep polyclonal anti-bovine prothrombin antibody (LSBio catalog LS-B10509) was added to 5 µL protein G Sepharose 4 Fast Flow beads (GE Healthcare catalog 17-0618-01) in water, incubated for 10 min with constant flipping. The antibody bound protein G beads were spun down and added to 1 mL cell culture media with constant flipping at 4 °C overnight. The prothrombin-bound protein G beads were then removed from the media by a quick spin, followed by passing the prothrombin-depleted media through a sterile filter. For infection, NHBE or HSAEC cells were plated in 384-wells at 2500 cells/well in either normal media, bovine prothrombin depleted with or without supplement of 1 µg/ml human prothrombin media for overnight. The cells were infected with omicron BA.2 strain of pSARS-2 at 6.69 × 10⁶ copies of RNA per well for 24 h before performing clotting assay as described above.

For component-based fibrin clotting assays, 100 ng of human prothrombin (Millipore, catalog 539515) was incubated with 100 ng factor Xa (R&D Systems, Inc. catalog 1063-SE-010) or 500 ng recombinant matriptase (R&D Systems, Inc. catalog 3946-SEB-010) or 200 ng of HAT (R&D Systems, Inc. catalog 2695-SE-010) in 40 µl volume in a 384-well plate at room temperature for one hour in 20 mM HEPES, 137 mM NaCl, 5 mM CaCl₂ prior to adding 1.5 µM fibrinogen to the mix. Upon addition of fibrinogen, the fibrin clotting was monitored with absorbance at 350 nm every 2 min on a plate reader.

## SEM sample preparation

Clotting assays were performed in a 24-well plate with 5 × 7 mm silicon chips (Ted Pella Inc., CA) immersed. Upon clotting, samples were fixed with 2% paraformaldehyde, and then post-fixed with 1.0% osmium tetroxide, and 0.8% potassium ferricyanide in 0.1 M sodium cacodylate buffer, stained with 1% tannic acid in dH₂O. After additional buffer washes, the samples were further osmicated with 2% osmium tetroxide in 0.1 M sodium cacodylate, then washed with dH₂O. Specimens were dehydrated with a graded ethanol series, critical point dried under CO₂

in a Bal-Tec model CPD 030 Drier (Balzers, Liechtenstein), mounted on aluminum studs, and sputter coated with 35 Å of iridium in a Quorum EMS300T D sputter coater (Electron Microscopy Sciences, Hatfield, PA) prior to viewing at 5 kV in a Hitachi SU-8000 field emission scanning electron microscope (Hitachi, Tokyo, Japan).

## Enzymatic cleavage of fluorogenic prothrombin and fibrinogen peptides

Both fluorogenic prothrombin and fibrinogen peptides were synthesized by Biomatik. The prothrombin peptide, Thrb-324, was synthesized as dabcyl-FNPRTFGSGE-edans corresponding to residue 324–333 of human prothrombin gene. The peptide encompasses the factor Xa cleavage site. The fluorogenic fibrinogen peptide, FPB, was synthesized as dabcyl-SARGHRPLE-edans, corresponding to amino acids 42–49 of human fibrinogen-β that includes thrombin cleavage site. The cleavage of Thrb-324 peptide was initiated by mixing 10 µM of the peptide with 100 ng of human factor Xa (R&D Systems, Inc), or 400 ng of human matriptase (R&D Systems, Inc) in 100 µl assay buffer containing 25 mM Tris at pH 9.0, 2.5 µM ZnCl₂, and 0.005% Brij-35 (w/v), or with infected cells or 100 µl of infected supernatant in 96-well plates. The cleavage of FPB peptide was carried out by mixing 10 µM of the peptide with 50 µl of infected or uninfected NHBE supernatants and 50 µl of the assay buffer in the presence or absence of 10 µM thrombin inhibitor dabigatran. The enzymatic cleavage reactions were monitored using a Synergy_H1 fluorescent plate reader (BioTek) with 340 nm excitation and 490 nm emission wavelengths for 3–4 h at 37 degrees.

## Western blot

NHBE cells were plated in 6-well plates and incubated at 37 °C, 5% CO₂ for 24 h. Cells were infected with SARS-CoV-2 pseudovirus for 24 h. Following infection, media was removed from cells and cells were washed with DPBS twice. Media was replaced with 50 mM HEPES, 250 mM NaCl buffer. Cells in buffer were incubated at 37 °C, 5% CO₂ for 0.5–1 h, then cells and supernatant were harvested. The cells were lysed with RIPA lysis buffer containing protease inhibitors. Proteins in supernatant were precipitated with 20% trichloroacetic acid (TCA) at 4 °C at least 10 min. The precipitated protein was spun down at 18,000 × g for 5 min, then washed two times with 200 µl cold acetone. The pellet was dried and then dissolved in SDS buffer for gel electrophoresis using NuPAGE 4–12% Bis-Tris gel. For western blot, proteins were transferred from the gel to PVDF membranes using iBlot transfer apparatus. The membrane was blocked with PBS containing 0.05% Tween-20 and 10% (w/v) skin milk for 1 h at room temperature (RT), then incubated with primary antibody (1:500 dilution of Anti *ST14*: A6135 from Abclonal, 1:1000 dilution of anit-*TMPRSS11D*: PA5-87660 from Invitrogen) for 1 h at RT or 4 degrees overnight. After three 5-min washes with blocking buffer, appropriate secondary antibodies at 1:2000 dilution were added for 1 hr at RT. Membrane was developed using SuperSignal West Dura Extended Duration Substrate (Thermo).

## Imaging of fibrin fibers by confocal microscopy

Fibrinogen was labeled with a fluorescent dye TAMRA-SE (Thermo Fisher Scientific, catalog c1171) according to the manufacture's protocol. The fluorescent TAMRA-fibrinogen was added to fibrin clotting assays at 80 µg/ml concentration or mixed with unlabeled fibrinogen at 1:6 ratio. Images were taken on a Zeiss LSM 880 confocal microscope equipped with Plan-Apochromat 20x/0.8 M27 objective. Z-stacks were performed to image fibrin formation. After acquisition, maximum intensity projections of the z-stacks were made using Fiji.

## Proteomics analyses by mass spectrometry

Twenty microliter aliquots of BALF samples were dissolved in SDS-sample buffer and applied onto a 4–12% Nupage gel with MOps running buffer. The run stopped after the samples migrated approximately ¼

distance into the gel. Each lane of the gel was sliced into smaller pieces, and subjected to destaining, reducing/alkylation, and in-gel trypsin digestion. The extracted peptides were applied for LC-MS/MS analysis using either a Thermo Orbitrap Fusion or a Thermo Orbitrap Fusion Lumos operated with an in-line Thermo nLC 1200 and an EASY-Spray ion source. Pepetides were separated using a 2 cm Pepmap 100 C18 trap column and a 25 cm Easy-spray Pepmap 100 C18 analytical column. MS/MS data acquisitions were operated at a 120,000 resolution ($m/z$ 200) with a scan range of 350–1950 $m/z$ and CID fragmentation. All data was processed using Proteome Discoverer v2.4 (Thermo Scientific) with a SEQUEST HT search against the Uniprot KB/Swiss-Prot Human Proteome (02/2021) and common contaminants (theGPM.org) using a 5 ppm precursor mass tolerance and a 0.5 Da fragment tolerance. Dynamic modifications included in the search were limited to oxidation [M], deamidation [NQ], and acetylation [Protein N-terminal] while carbamidomethylation [C] was the only static modification utilized. Peptides and proteins were filtered at a 1% FDR (False Discovery Rate) using a target-decoy approach with a 2 peptide per protein minimum. As *ST14* and *TMPRSS11D* are members of glycosylated cell surface transmembrane proteins, they present difficulty in peptide analysis due to poor fragmentation and glycan heterogeneity. Thus, a less stringent FDR was applied to identify *ST14* and *TMPRSS11D* peptides in COVID BALF. Relative protein abundance was estimated from an average of its top 3 unique peptide intensities as determined by chromatographic area-under-the-curve and normalized by total intensity of all peptides. The sum of abundances in a dataset is further normalized to 1,000,000. Pearson correlation coefficients between samples were calculated using normalized abundance of each protein with exclusion of serum albumin and immunoglobulin genes, whose abundances are donor-dependent. The differential abundance is calculated as percentage of difference in abundance: by dividing the difference abundance between a protein in one sample and the average abundance of the protein with the average abundance of the protein from all healthy samples. The list of proteins used for the differential abundance heatmap analysis include the ones with average healthy abundance greater than 25 and all none zero abundance in the acute COVID sample. The heatmaps display the differential abundance relative to the average of each protein.

### Fibrinogen, prothrombin, and IgG ELISA
ELISA assays were used to determine the levels of fibrinogen (Abcam, ab108841), total IgG (Abcam, ab195215), and prothrombin (Molecular Innovations, HPTKT-TOT) present in human BALF samples. The samples were diluted with kit specific assay diluents. For prothrombin and fibrinogen levels, samples were evaluated at 1:50 and 1:500 dilutions. For the total IgG ELISA, samples were evaluated at 1:1000 and 1:10,000 dilutions. The assays were carried out following the manufacture's protocols.

### RNAseq sample preparation
Total RNA was extracted from ~1 × 10⁶ NHBE or HSAEC cells with Trizol (Invitrogen, Carlsbad, CA, USA) according to manufacturer's instructions. Ten μg of purified RNA from each sample was sent to Genewiz commercial sequencing facility (South Plainfield, NJ) for Bioanalyzer quality control analysis (Agilent, Santa Clara, CA) and Illumina Next Generation Sequencing. All the submitted total RNA samples had an RNA integrity number (RIN) of 10.

### CRISPR/Cas9 knockdown of tissue factor gene in NHBE and HSAEC cells
Tissue factor gene knockdown on NHBE and HSAEC cells were performed using CRISPR/Cas9 ribonucleoprotein (RNP) complex delivery method[48,76]. Briefly, Alt-R® S.p. Cas9 Nuclease V3, target

specific CRISPR RNA (crRNA), target-independent trans-activating CRISPR RNA (tracrRNA), and Duplex Buffer were purchased from Integrated DNA Technologies, Inc (Coralville, IA). The sequence of the tissue factor crRNA: TTCCCTCCCGAACAGTTAAC, was synthesized by Integrated DNA Technologies, Inc. Alt-R® Cas9 Negative Control crRNA was used as a negative control. Both tracrRNA and crRNA were resuspended in Duplex Buffer to 100 μM and then annealed to a final concentration of 10 μM in the same buffer. A 1.4 μl aliquot of AltR S.p. Cas9 was added to 7 μl of PBS, and mixed with 16.8 μl of crRNA-tracrRNA hybrid. The mixture was diluted with 210 μl Opti-MEM I medium, mixed with 15 μl Plus Reagent (Thermo Fisher Scientific) for 10 min to form the RNP complex solution. Separately, 10 μl lipofectamine CRISPRMAX (Thermo Fisher Scientific) was diluted with 240 μl Opti-MEM, and then mixed with the RNP solution and incubated for a 7 min at room temperature. For transfection, the lipofectamine and RNP mixture was added to 2 × 10⁵ NHBE or HSAEC cells suspended in 5 ml respective culture media. Media was exchanged every two days post transfection. In about 10 days after transfection, cells were lysed using RIPA lysis buffer containing protease inhibitors and western blot was performed to verify tissue factor expression. The primary and secondary antibodies used were polyclonal goat anti-human tissue factor antibody (R&D Systems, inc. Catalog AF2339) at 1:400 dilution and HRP-conjugated rabbit anti-Goat IgG secondary antibody at 1:2000 dilution (R&D Systems, Minneapolis, MN), respectively.

### Statistics and reproducibility
All clotting graphs are representative of duplicate wells from one of 2 or 3 independent experiments. All confocal, EM, and western blot images are representative of at least 2 independent experiments. No statistical method was used to predetermine sample size. The experiments were not randomized. The investigators were not blinded to allocation during experiments and outcome assessment.

## Data availability
The RNA-Seq data generated in this study have been deposited in NCBI GEO database (https://www.ncbi.nlm.nih.gov/) under accession code GSE242722. The mass spectrometry proteomics data have been deposited to the ProteomeXchange Consortium via the PRIDE (http://www.ebi.ac.uk/pride) partner repository[77]. The dataset can be accessed with identifier PXD045119. All other data supporting the findings of this study are available within the paper and its Supplementary Information. Source data are provided with this paper in the Source data file. Source data are provided with this paper.

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

## Acknowledgements

We thank Dr. Glenn Nardone for helping the protein ID part of the project. The work is supported by a COVID funding project AI001263 for P.D.S. from the division of intramural research, National Institute of Allergy and Infectious Diseases, National Institutes of Health.

## Author contributions

R.E., C.H., C.A., J.I., G.R., Z.Z., J.L., B.A.P.L., N.L.G., B.B., M.Z., M.S., E.R.F., and R.F.J. performed the experimental studies, L.O., J.B., E.R.F., and R.F.J. carried out the analyses, H.T. provided clinical samples, P.D.S. conceived the project, supervised the work, and wrote the manuscript.

## Funding

## Competing interests

The authors declare no competing interests.
