## [Peer Review File · Nature Communications]

Reviewers' Comments:

Reviewer #1:

Remarks to the Author:

Erickson et al present a new mechanistic model for COVID-19 induced lung fibrosis. The mechanism of SARS-CoV-2-induced lung fibrosis is not well understood. The authors propose infection-activated shedding of lung epithelial cell transmembrane proteases, such as ST14 and TMPRSS11D. Importantly, their data showed that infected but not uninfected NHBE cells produced fibrin clots in the presence of bronchoalveolar lavage fluid (BALF) derived from acute COVID patients but not healthy individuals. If correct, their proposed mechanistic model may reveal the inefficiency of current heparin-based therapy as the infected cell-mediated fibrosis occurs in alveoli outside of blood circulation and suggest the need to treat respiratory airway rather than blood vessels with direct thrombin inhibitors. Identifying the mechanism by which SARS-CoV-2 induces fibrin clot formation and lung fibrosis is important in developing new therapeutic treatments to reduce mortality associated with the ongoing COVID-19 pandemic. The experiments are generally sound, but there are a few concerns:

Major:

1. The pseudotyped viruses were made by pseudotyping the SARS-CoV-2 spike protein S onto the HIV NL4-3 env-nef-luciferase core virus. The data in all subfigures within figures 2-5 where these pSARS-2 pseudotypes are used (from any of the viral strains) is missing a very important control: bald virus. This is a standard control in any pseudotyped virus studies, which is very important to show that the effects observed are due to the SARS-CoV-2 S proteins only, and not to some other protein or factor in the HIV NL4-3 vector, such as the env, nef, gag, luciferase, etc... proteins, or even some other cellular factor inadvertently incorporated into the pSARS-2 virions. In these subfigures only a uninfected control was used, which does not consider the effects of any of the said proteins or cellular factors.
2. The bald virus described in Major point # 1 needs to be added at the same genome copies levels as those used for pSARS-2 virions, assayed and determined in the same manners.
3. The precise mechanism of pulmonary fibrosis is not well described. A more detailed description of what exact BALF factors are key to inducing, allowing, and inhibiting such fibrosis (such as from those in Supplementary Figure 1 and Table S1) could be relatively easily studied using the assays the authors have already set up.
4. Also on the mechanism theme, how the SARS-CoV-2 Spike protein (S) induces lung fibrosis is not really addressed in the study. What regions of the protein are important and why?
5. The Discussion section lacks detail, which may be derived from the mechanistic studies described in Major Point # 3 above.

Relatively Minor:

1. For Figure 7, the description of the results focuses on BALF from R3232, but not R3248, even though they present similar phenotypes. They should be described together in the Results section. Same applies to BALF from C3263 and C3267.

Reviewer #2:

Remarks to the Author:

This MS by Erickson et al applies in vitro studies to investigate the roles of thrombin and epithelial cell derived serine proteases in COVID related lung fibrosis. They report lung epithelial cells infected with COVID shed of transmembrane proteases, such as ST14 or TMPRSS11D which activate thrombin to generate fibrin fibers, that these fibrin fibers are profibrotic. They also report that prothrombin and fibrinogen levels are elevated in BALF samples from acute COVID patients and that fibrin clots were only observed in the COVID BALF. The authors conclude that the findings

reveal the inefficiency of heparin-based therapy for COVID fibrosis is the infected cell-mediated fibrosis occurs in alveoli outside of blood circulation and suggest the airways need to be targeted with direct thrombin inhibitors.

Major

The MS is framed around the phenomenon of COVID fibrosis. However, the biology outlined in the introduction is largely the pathobiology of acute lung injury and misuses the term fibrosis. For example, the introduction cites 4 early reports of COVID lung "fibrosis". However, those reports did not document durable fibrosis. Rather they reported changes consistent with organizing acute lung injury. The citations should be edited to outline these points. Also, the introduction also discusses the failed use of heparin for COVID fibrosis. Heparin was used in the acute phases of disease, not fibrotic phase, thus it's failure is unrelated to fibrosis.

Overall, the studies in this MS likely relate to the pathobiology of acute lung injury/organizing diffuse alveolar damage. Active COVID infection causes DAD. The rare (fibrosis develops in no more than 4% of severe COVID cases) cases of COVID fibrosis occur late, at least 6 months or more after COVID infection- well after the virus is cleared. Thus, experiments using virus likely relate to biologic processes that occur early in disease, not the later fibrotic stage of disease. The return of fibrinogen levels to normal in COVID-recovered BALF are consistent with these conclusions.

The patient-based studies are small. For example, the proteomics sample size is very small (only 1 acute/1 recovered disease sample). Given small sample size, it is uncertain if the findings are representative of the disease process.

The data are dependent on bovine thrombin in culture media. This raises questions of whether the findings are relevant to humans. Concerns include whether human serine proteases would have the same effects on human thrombin, whether the relative concentrations of proteases and thrombin in the culture system correlate to what is found in patients, or whether the same processes would occur in vivo when protease inhibitors are also present in their relevant concentrations.

No data are provided supporting the hypothesis that the fibrin fibers are profibrotic.

Reviewer #3:

Remarks to the Author:

The manuscript by Erickson et al examines the cellular/molecular basis for extravascular fibrin deposits that occur in the lungs of SARs-CoV-2 infected individuals. The authors first use proteomics to identify proteins that accumulate in the BAL fluid of COVID patients and observe increased levels of a subset of coagulation proteins (e.g., fibrinogen, prothrombin, FXIII) in acute COVID but not healthy or recovered-COVID patients. The authors then speculate that a unique cell-based mechanism of clotting may be responsible for fibrin deposits in the airways in COVID. In support of this concept, the authors establish a cell-based assay with normal human brachial/tracheal epithelial cells (NHBE). Here, the authors infect NHBE cells with SARs virus and show that these cells acquire the ability to support fibrin clot formation in vitro. Additional data suggests that the SARs-infected NHBE cells promote fibrin formation through a mechanism dependent on (pro)thrombin. Finally, the authors attempt to show that fibrin formation is independent of Factor X, but dependent on TMPRSS family member serine proteases. The study is significant and the concept that unique mechanisms of clotting may occur in COVID is intriguing and innovative. However, there are several significant flaws, including promoting the concept that extravascular fibrin deposition is equivalent to fibrosis, a naivete regarding how the coagulation system functions, missing experiments and controls, and conclusions not supported by the data.

Major comments:

- The authors equate extravascular fibrin deposition with fibrosis. These are not the same

processes. Fibrin is the product of coagulation system activation and clotting. Fibrosis is driven by myofibroblast activation, epithelial cell hyperplasia, and collagen deposition (and other matrix proteins) that ultimately leads to scar formation in tissues. Indeed, several studies of lung injury in animal models have shown that fibrosis occurs in the lungs even in the complete absence of fibrinogen (Hattori et al., JCI 2000; Ploplis et al., Am J Path 2000, Wilberding et al., Ann NT Acad Sci 2001). To equate fibrin formation with fibrosis is simply incorrect.

- The authors show that SARs-infected NHBE cells can support fibrin formation in vitro. In their model, this apparently requires (pro)thrombin coming from the fetal bovine serum used to culture cells. The authors speculate that the thrombin is activated by proteases associated with or released from the infected NHBE cells. However, the authors appropriately note that SARs-infection of the NHBE cells can result in significant upregulation of tissue factor. Did the authors document this infection-mediated upregulation of TF? More importantly, the authors should have performed experiments inhibiting TF (e.g., with inhibitory antibodies) or suppressing TF expression (e.g., siRNA) to confirm that the source of thrombin generation in this system is not TF-dependent. These experiments are essential to support the authors conclusion and proposed mechanism.

- In addition to the prothrombin carry over from fetal bovine serum used to culture the cells, it is possible that other coagulation factors (e.g., FX, FV, FVII) are also carried over, and that it is actually Xa-mediated thrombin generation being observed. Did the authors perform experiments specifically inhibiting Xa (e.g., with rivaroxaban) as they did with the experiments inhibiting thrombin (e.g., with dabigatran)? These experiments are essential to support the authors conclusion and proposed mechanism.

- The authors provide data showing that recombinant matriptase and HAI can cleave a prothrombin peptide. However, do the authors have data showing that matriptase and HAI can produce catalytically active thrombin, e.g., using a thrombin generation assay? Do the authors have data showing infected NHBE cell supernatant can support thrombin generation, specifically in the presence of Xa inhibitors?

- On page 7 the authors state: "we adopted a turbidity-based fibrin clotting assay to measure fibrin aggregation resulting from cleavage of fibrinogen peptides." Thrombin-mediated cleavage of fibrinogen does not initiate aggregation of fibrin, it initiates polymerization of fibrin.

- On page 13 the authors state: "Indeed, anti-coagulation factors, such as plasminogen..." Plasminogen is not an anti-coagulation factor. It is a fibrinolytic factor.

- In Figure 7, the authors perform functional studies of fibrin formation with BALF used for the proteomics analyses. Based on the cell culture studies, can the authors document the presence of matriptase or HAI in these fluids, specifically the fluids collected from acute COVID patients? Does this fluid alone support thrombin generation? Is it inhibitable with BB-94 or prinomastat?

- A significant failure of this manuscript is lack of documentation that the proposed mechanism actually occurs in vivo (using an animal model system or otherwise). Indeed, the authors have in no way excluded the possibility that the canonical extrinsic or intrinsic coagulation pathways are responsible (whole in part) for extravascular fibrin deposits in COVID. Moreover, no data is provided to support that the matriptase/HAI-mediated activation of prothrombin occurs in vivo.

We thank all three reviewers for their in-depth review of our manuscript and for their generally supporting comments. We appreciate the reviewer's statement about our work as: "*The study is significant and the concept that unique mechanisms of clotting may occur in COVID is intriguing and innovative.*". The reviewers also requested further experiments to address mechanistic questions, such as knocking down of tissue factors in the primary lung epithelial cells to address the contribution of classical coagulation pathway to SARS-CoV-2 infection induced fibrin clotting. The reviewers also requested us to distinguish fibrosis with acute pulmonary damage. We have performed all the suggested experiments and extensively revised the manuscript to focus on SARS-CoV-2 infection associated acute DAD-related lung damage. Below is a detailed response to the reviewers comments.

Reviewers' comments:

Reviewer #1 (Remarks to the Author):

Erickson et al present a new mechanistic model for COVID-19 induced lung fibrosis. The mechanism of SARS-CoV-2-induced lung fibrosis is not well understood. The authors propose infection-activated shedding of lung epithelial cell transmembrane proteases, such as ST14 and TMPRSS11D. Importantly, their data showed that infected but not uninfected NHBE cells produced fibrin clots in the presence of bronchoalveolar lavage fluid (BALF) derived from acute COVID patients but not healthy individuals. If correct, their proposed mechanistic model may reveal the inefficiency of current heparin-based therapy as the infected cell-mediated fibrosis occurs in alveoli outside of blood circulation and suggest the need to treat respiratory airway rather than blood vessels with direct thrombin inhibitors. Identifying the mechanism by which SARS-CoV-2 induces fibrin clot formation and lung fibrosis is important in developing new therapeutic treatments to reduce mortality associated with the ongoing

COVID-19 pandemic. The experiments are generally sound, but there are a few concerns:

Major:

- 1. The pseudotyped viruses were made by pseudotyping the SARS-CoV-2 spike protein S onto the HIV NL4-3 env-nef-luciferase core virus. The data in all subfigures within figures 2-5 where these pSARS-2 pseudotypes are used (from any of the viral strains) is missing a very important control: bald virus. This is a standard control in any pseudotyped virus studies, which is very important to show that the effects observed are due to the SARS-CoV-2 S proteins only, and not to some other protein or factor in the HIV NL4-3 vector, such as the env, nef, gag, luciferase, etc... proteins, or even some other cellular factor inadvertently incorporated into the pSARS-2 virions. In these sub-figures only a uninfected control was used, which does not consider the effects of any of the said proteins or cellular factors.*
- 2. The bald virus described in Major point # 1 needs to be added at the same genome copies levels as those used for pSARS-2 virions, assayed and determined in the same manners.*

Response: We thank the reviewer for providing generally positive view of our manuscript. We have included results of infection-induced fibrin clotting using both pSARS-2 and env-null bald virus in the same dose titration range of viral RNA copies in Supplemental Figure 2G. While the infection of NHBE cells with SARS-CoV-2 spike containing pseudoviruses resulted in a dose-dependent fibrin clotting, the infection with bald viruses did not result in fibrin clotting in the

same dose range. These results show that the spike protein is needed for the infection-induced fibrin clot formation and the spike-negative bald virus failed to induce fibrin clot formation. It is likely that the infection by bald virus is much lower than that of spike-containing virus.

3. The precise mechanism of pulmonary fibrosis is not well described. A more detailed description of what exact BALF factors are key to inducing, allowing, and inhibiting such fibrosis (such as from those in Supplementary Figure 1 and Table S1) could be relatively easily studied using the assays the authors have already set up.

4. Also on the mechanism theme, how the SARS-CoV-2 Spike protein (S) induces lung fibrosis is not really addressed in the study. What regions of the protein are important and why?

5. The Discussion section lacks detail, which may be derived from the mechanistic studies described in Major Point # 3 above.

Response: We have provided a better description of the SARS-CoV-2 infection-induced fibrin clotting mechanism in the revised discussion section of the manuscript. In this model, SARS-CoV-2 infection of lung epithelial cells results in the activation of type II transmembrane serine proteases (TMPRSS) family members. The activated TMPRSS genes possess factor X-like activities to cleave prothrombin into activated thrombin, which in turn cleaves fibrinogen resulting in fibrin deposition in infected alveolar space (Supplemental Figure 7B). Importantly, this infection-induced fibrin clotting occurs in the absence of tissue factor expression and do not require other serine proteases in plasma coagulation pathways. To address how the SARS-CoV-2 spike protein induces fibrin clotting, we first asked if the infection-induced fibrin clotting is unique to SARS-CoV-2 infection. To this end, we compared fibrin clot formation induced by SARS-CoV-2 and VSV-pseudoviruses infected NHBE cells and observed fibrin clot formation in VSV-pseudoviruses infected NHBE cells, albeit with less magnitude than that of pSARS-2 (supplemental Figure 2G). These results suggest the infection-induced fibrin clotting is not unique to SARS-CoV-2 virus, and infection by other viruses may also induce fibrin deposition.

Relatively Minor:

1. For Figure 7, the description of the results focuses on BALF from R3232, but not R3248, even though they present similar phenotypes. They should be described together in the Results section. Same applies to BALF from C3263 and C3267.

Response: We agree with the reviewer that R3232 and R3248 both clotted in the presence of the viral infection. The aggregates observed in UI C3267, however, lacked fibrin polymer morphology compared to UI C3263. We have revised the manuscript accordingly.

Reviewer #2 (Remarks to the Author):

This MS by Erickson et al applies in vitro studies to investigate the roles of thrombin and epithelial cell derived serine proteases in COVID related lung fibrosis. They report lung epithelial cells infected with COVID shed of transmembrane proteases, such as ST14 or TMPRSS11D which activate thrombin to generate fibrin fibers, that these fibrin fibers are profibrotic. They also report that prothrombin and fibrinogen levels are elevated in BALF samples from acute COVID patients and that fibrin clots were only observed in the COVID BALF. The authors conclude that the findings reveal the inefficiency of heparin-based therapy

for COVID fibrosis is the infected cell-mediated fibrosis occurs in alveoli outside of blood circulation and suggest the airways need to be targeted with direct thrombin inhibitors.

Major

The MS is framed around the phenomenon of COVID fibrosis. However, the biology outlined in the introduction is largely the pathobiology of acute lung injury and misuses the term fibrosis. For example, the introduction cites 4 early reports of COVID lung “fibrosis”. However, those reports did not document durable fibrosis. Rather they reported changes consistent with organizing acute lung injury. The citations should be edited to outline these points. Also, the introduction also discusses the failed use of heparin for COVID fibrosis. Heparin was used in the acute phases of disease, not fibrotic phase, thus it’s failure is unrelated to fibrosis.

Overall, the studies in this MS likely relate to the pathobiology of acute lung injury/organizing diffuse alveolar damage. Active COVID infection causes DAD. The rare (fibrosis develops in no more than 4% of severe COVID cases) cases of COVID fibrosis occur late, at least 6 months or more after COVID infection- well after the virus is cleared. Thus, experiments using virus likely relate to biologic processes that occur early in disease, not the later fibrotic stage of disease. The return of fibrinogen levels to normal in COVID-recovered BALF are consistent with these conclusions.

Response: We thank the reviewer for pointing out the distinction between fibrosis and DAD related acute lung damage. We have revised the manuscript to extensively to focus on the description of DAD related acute lung damage rather than fibrosis.

The patient-based studies are small. For example, the proteomics sample size is very small (only 1 acute/1 recovered disease sample). Given small sample size, it is uncertain if the findings are representative of the disease process.

Response: We have expanded our proteomic studies to include all four acute COVID, four recovered COVID and three healthy BALF samples. The results are included in revised Figure 1A-D and supplemental Figure 1A-E. Consistently, we observed enrichment of complement proteins, some coagulation proteins in acute but not recovered nor healthy BALF. While the overall conclusion remains the same as before, the increased sample number now allow us to run statistical analyses on each category proteins between different COVID status.

The data are dependent on bovine thrombin in culture media. This raises questions of whether the findings are relevant to humans. Concerns include whether human serine proteases would have the same effects on human thrombin, whether the relative concentrations of proteases and thrombin in the culture system correlate to what is found in patients, or whether the same processes would occur in vivo when protease inhibitors are also present in their relevant concentrations.

Response: To address the bovine vs human thrombin issue. We depleted bovine prothrombin from NHBE culture media and then supplemented human prothrombin in the media. Depletion of bovine prothrombin resulted in a loss in fibrin clot formation in the presence of pSARS-2 infection. Importantly, adding human prothrombin at 1ug/ml restored fibrin clotting ability of

infected NHBE cells. These results showed that indeed human serine proteases activated human prothrombin similar to the bovine homolog. The concentrations of experimental prothrombin and fibrinogen are relevant to those found in patients based on the following: first, the prothrombin concentration in acute COVID BALF is between 0.2-3ug/ml by ELISA (Figure 1E). As the concentrations of proteins in BALF are likely ~20 fold diluted from their concentrations in lung epithelial lining fluid, suggesting the prothrombin concentration in acute COVID lung is close to ~2-20ug/ml in patients. We estimated the bovine prothrombin in cell culture media to be less than 0.5ug/ml and we used 1ug/ml human prothrombin both in supplemented culture media and in the component clotting assays (Figure 6B). Second, fibrin clot formed in pSARS-2 infected NHBE cells with the addition of acute COVID BALF (Figure 7) without additional prothrombin nor fibrinogen. Figure 7 demonstrated that the concentrations of prothrombin and fibrinogen in acute COVID patient lung is sufficient to drive fibrin clotting in the presence of the viral infection. As to whether the fibrin clotting occurs in vivo when protease inhibitors are present in their relevant concentrations, we think this is the key finding of this work. Namely, fibrin clotting would still occur in SARS-CoV-2 infected lungs in current hospital delivery of anticoagulant by either IV or oral, both are effective in inhibiting plasma fibrin clotting in circulation. As the concentration of systemically injected inhibitors in lung is dependent on diffusion from circulation to lung, thus is expected to be lower than in plasma. In addition, the mechanism of SARS-CoV-2 infection-induced fibrin clotting in acute COVID lung BALF does not depend on plasma tissue factor and other coagulation enzymes, suggesting it would be difficult to increase the inhibitor concentration in plasma without risking bleeding. On the other hand, direct application of the inhibitors to lung space would allow effective inhibition of fibrin clotting without risking bleeding.

No data are provided supporting the hypothesis that the fibrin fibers are profibrotic.

Response: we have revised manuscript as suggested by the reviewer to focus on DAD-related acute fibrin deposition resulted from SARS-CoV-2 infection. We are not addressing fibrosis.

Reviewer #3 (Remarks to the Author):

The manuscript by Erickson et al examines the cellular/molecular basis for extravascular fibrin deposits that occur in the lungs of SARS-CoV-2 infected individuals. The authors first use proteomics to identify proteins that accumulate in the BAL fluid of COVID patients and observe increased levels of a subset of coagulation proteins (e.g., fibrinogen, prothrombin, FXIII) in acute COVID but not healthy or recovered-COVID patients. The authors then speculate that a unique cell-based mechanism of clotting may be responsible for fibrin deposits in the airways in COVID. In support of this concept, the authors establish a cell-based assay with normal human brachial/tracheal epithelial cells (NHBE). Here, the authors infect NHBE cells with SARS virus and show that these cells acquire the ability to support fibrin clot formation in vitro. Additional data suggests that the SARS-infected NHBE cells promote fibrin formation through a mechanism dependent on (pro)thrombin. Finally, the authors attempt to show that fibrin formation is independent of Factor X, but dependent on TMPRSS family member serine proteases. The study is significant and the concept that unique mechanisms of clotting may occur in COVID is intriguing and innovative. However, there are several significant flaws, including promoting the concept that extravascular fibrin deposition is equivalent to fibrosis, a naivete regarding how the coagulation system functions, missing

experiments and controls, and conclusions not supported by the data.

Major comments:

- The authors equate extravascular fibrin deposition with fibrosis. These are not the same processes. Fibrin is the product of coagulation system activation and clotting. Fibrosis is driven by myofibroblast activation, epithelial cell hyperplasia, and collagen deposition (and other matrix proteins) that ultimately leads to scar formation in tissues. Indeed, several studies of lung injury in animal models have shown that fibrosis occurs in the lungs even in the complete absence of fibrinogen (Hattori et al., JCI 2000; Ploplis et al., Am J Path 2000, Wilberding et al., Ann NT Acad Sci 2001). To equate fibrin formation with fibrosis is simply incorrect.

Response: We agree with the reviewer that fibrosis and DAD-related fibrin deposition are two different process and have revised manuscript extensively to focus on fibrin deposition not lung fibrosis.

- The authors show that SARs-infected NHBE cells can support fibrin formation in vitro. In their model, this apparently requires (pro)thrombin coming from the fetal bovine serum used to culture cells. The authors speculate that the thrombin is activated by proteases associated with or released from the infected NHBE cells. However, the authors appropriately note that SARs-infection of the NHBE cells can result in significant upregulation of tissue factor. Did the authors document this infection-mediated upregulation of TF? More importantly, the authors should have performed experiments inhibiting TF (e.g., with inhibitory antibodies) or suppressing TF expression (e.g., siRNA) to confirm that the source of thrombin generation in this system is not TF-dependent. These experiments are essential to support the authors conclusion and proposed mechanism.

Response: We agree with the reviewer that we could not exclude the contribution of tissue factor-initiated coagulation in our previous submission. To address the involvement of TF, we performed RNAseq on pSARS-2 infected NHBE and HSAEC cells as well as on replication competent field strain delta variant SARS-CoV-2 virus infected NHBE cells. Both infected NHBE and HSAEC cells exhibited upregulation of many antiviral and interferon-regulated genes (Figure 5C). However, we did not observe significant upregulation in the expression of TF in neither infected NHBE and nor infected HSAEC cells (Figure 5C). As both cells express TF (detectable on westernblots), we then performed knockdown of TF in both NHBE and HSAEC cells using CRISPR/cas9 method (Figure 6D). Importantly, pSARS-2 infected but not uninfected TF KD NHBE and HSAEC cells induced fibrin clotting comparable to those by wildtype control NHBE and HSAEC cells (Figure 6E), suggesting SARS-CoV-2 infection-induced fibrin clotting is independent of tissue factor.

- In addition to the prothrombin carry over from fetal bovine serum used to culture the cells, it is possible that other coagulation factors (e.g., FX, FV, FVII) are also carried over, and that it is actually Xa-mediated thrombin generation being observed. Did the authors perform experiments specifically inhibiting Xa (e.g., with rivaroxaban) as they did with the experiments inhibiting thrombin (e.g., with dabigatran)? These experiments are essential to support the authors conclusion and proposed mechanism.

Response: We did use rivaroxaban to inhibit the fibrin clot formation. However, the compound also potentially inhibited thrombin in the positive control that consisted of only thrombin and fibrinogen. Namely, the inhibitory activity of rivaroxaban in our infection-induced fibrin clotting assay is likely due to the compound inhibition of thrombin. It is possible that factor Xa-mediated thrombin activation contributed to low level fibrin clotting observed in the uninfected cells in the presence of acute COVID BALF C3263 (Figure 7B), and the SARS-CoV-2 infections induced tissue factor and factor Xa independent fibrin clotting (Figure 6E, 7B).

- The authors provide data showing that recombinant matriptase and HAI can cleave a prothrombin peptide. However, do the authors have data showing that matriptase and HAI can produce catalytically active thrombin, e.g., using a thrombin generation assay? Do the authors have data showing infected NHBE cell supernatant can support thrombin generation, specifically in the presence of Xa inhibitors?

Response: Yes. Not only matriptase and HAT (TMPRSS11D) cleaved prothrombin peptide (Figure 6A), both promoted fibrin clot formation in our in vitro assay consisting of matriptase+ prothrombin + fibrinogen or HAT+ prothrombin+fibrinogen (Figure 6B). Prothrombin alone with fibrinogen only resulted in low level or not detectable fibrin clot formation. To address if infected NHBE cell supernatants support thrombin generation, we performed a thrombin generation assay using a fluorogenic fibrinogen peptide. Only active but not prothrombin cleaved the fibrinogen peptide. Importantly, infected but not uninfected NHBE supernatants cleaved the fibrinogen peptide (Supplemental Figure 5A), and the cleavage can be inhibited by a thrombin inhibitor dabigatran.

- On page 7 the authors state: “we adopted a turbidity-based fibrin clotting assay to measure fibrin aggregation resulting from cleavage of fibrinogen peptides.” Thrombin-mediated cleavage of fibrinogen does not initiate aggregation of fibrin, it initiates polymerization of fibrin.

Response: we changed wording from aggregation to polymerization. We agree with the reviewer that thrombin cleavage initiate fibrin polymerization. It is the aggregation of the polymerized fibrin filaments resulted in a turbidity change for clot detection.

- On page 13 the authors state: “Indeed, anti-coagulation factors, such as plasminogen...” Plasminogen is not an anti-coagulation factor. It is a fibrinolytic factor.

Response: we changed wording and the revised manuscript states: “suggesting fibrinogen may not be the only factor controlling fibrin formation, and there may be other fibrinolytic factors, such as plasminogen, present in BALF to suppress fibrin polymerization. Other anti-coagulation factors, such as antithrombin-III and serine protease inhibitors (SERPIN) were also found in both healthy and COVID BALF (Supplemental Table 1)”

- In Figure 7, the authors perform functional studies of fibrin formation with BALF used for the proteomics analyses. Based on the cell culture studies, can the authors document the

presence of matriptase or HAI in these fluids, specifically the fluids collected from acute COVID patients? Does this fluid alone support thrombin generation? Is it inhibitable with BB-94 or prinomastat?

Response: We could not detect the presence of matriptase and HAT in any of the acute COVID BALF. However, low level of fibrin polymers were clearly visible in one of the acute sample, C3263 (Figure 7B), suggesting the BALF alone occasional support fibrin clotting. Since none of the healthy nor recovered BALF supported fibrin clotting without the viral infection, we think the activated thrombin in C3263 maybe generated during the SARS-CoV-2 infection in the patient lung. Nevertheless, infections dramatically enhanced fibrin clotting in these samples. We revised discussion section to address the comment.

- A significant failure of this manuscript is lack of documentation that the proposed mechanism actually occurs in vivo (using an animal model system or otherwise). Indeed, the authors have in no way excluded the possibility that the canonical extrinsic or intrinsic coagulation pathways are responsible (whole in part) for extravascular fibrin deposits in COVID. Moreover, no data is provided to support that the matriptase/HAI-mediated activation of prothrombin occurs in vivo.

Response: There is no good animal models existed to date for COVID-associated acute lung pathology. Three different animal models have been used extensively for COVID research: ACE2-transgenic mice, Syrian hamsters, and rhesus macaque (Jiang et al. Cell 182:50-58, 2020. <https://doi.org/10.1016/j.cell.2020.05.027>) (Rosenke et al. Emerging Microbes & Infections vol 9, 2020 <https://doi.org/10.1080/22221751.2020.1858177>) (Munster et al. Nature 585:268-272, 2020 <https://doi.org/10.1038/s41586-020-2324-7>). All can be infected by SARS-CoV-2 viruses. The infected animals often developed pneumonia and weight loss but then recovered in two weeks. Infected ACE2-transgenic mice die due to overtly immune response rather than DAD-related lung pathology. They rarely develop DAD or fibrin-rich hyaline membranes that are similar to human lung pathology. They are good for antiviral development but not for lung pathology. A review of using nonhuman primate as SARS-CoV-2 model stated: “none of the primates exhibited severe lesions or evidence of diffuse alveolar damage and therefore are unlikely to represent the severe form of SARS-CoV-2 infection observed in fatal human cases”(Clancy et al. ACVP volume 59, 2022 <https://doi.org/10.1177/03009858211067468>). We have consulted with labs using Syrian hamster and macaque as COVID models and both concluded animals lack the COVID-associated acute DAD-related lung pathology that observed in severe human COVID cases.

Reviewers' Comments:

Reviewer #1:

Remarks to the Author:

Erickson et al have presented a new mechanistic model for COVID-19 induced lung fibrosis. In this model, the spike S protein of SARS-CoV-2 induces fibrin clot formation. This is definitely an important and timely topic of research. The authors have done a fair job addressing my prior concerns. However, the following concern has not been appropriately addressed:

4. Also on the mechanism theme, how the SARS-CoV-2 Spike protein (S) induces lung fibrosis is not really addressed in the study. What regions of the protein are important and why?

While the first part of this important point was addressed, the second part: "What regions of the protein are important and why?" has not been addressed experimentally. Even very rough stalk vs. head vs. cytoplasmic/transmembrane regions could be addressed with not too much effort, for example by adding soluble S vs S stalk protein vs S head protein. Such an experiment would also allow for the clotting phenotype observed to be most definitive.

Reviewer #2:

Remarks to the Author:

The authors have provided comprehensive and adequate responses to my queries. There are no additional comments

Reviewer #3:

Remarks to the Author:

The authors were partially responsive to the questions and concerns raised during the initial review of the manuscript. Important additional data were provided using the NHBE cell culture system. Most notably the authors provide data suggesting TF does not contribute to fibrin formation in this system. There is now also data using human prothrombin that is compatible with the proposed mechanism. It is clear that there is prothrombin, fibrinogen, and other coagulation proteins in BALF fluid from infected individuals and that this material has the capacity to form fibrin. However, a major weakness of the manuscript remains. Namely, there is no data to suggest that the proposed mechanism of TTSP mediated activation of prothrombin occurs in the airway in vivo, as proposed in Supplement Figure 7. The authors could not document the presence of TTSP proteins in patient BALF (infected or non-infected). Moreover, it appears that there is not an animal model system amenable to testing the hypothesis. Finally, the last paragraph of the discussion accurately notes that heparins, which primarily target factor Xa but also thrombin, have not been beneficial in mitigating COVID associated acute lung damage. Here the authors suggest that this lack of efficacy is due to TTSP proteins bypassing factor X to activate thrombin. This entire argument is predicated on the concept that fibrin deposition in the alveolar space exacerbates acute lung injury following SARS-CoV2 infection. Whereas this may well be the case, this reviewer is not aware of any data definitively demonstrating this to be true. The fact that animal models are not amenable to testing the author's hypothesis put forth in the present study also precludes direct studies demonstrating a causative role for fibrin in driving DAD. At minimum, each of these weaknesses should be better addressed in the Discussion section of the manuscript.

We thank the reviewers for their in-depth review of our manuscript and for their constructing comments. We have addressed their comments through new experiments and through revising the manuscript. The revised contents are highlighted in red in the manuscript. Below is a point-by-point response to the reviewers comments with our responses highlighted in red.

Comment from Reviewer #1 (Remarks to the Author):

Erickson et al have presented a new mechanistic model for COVID-19 induced lung fibrosis. In this model, the spike S protein of SARS-CoV-2 induces fibrin clot formation. This is definitely an important and timely topic of research. The authors have done a fair job addressing my prior concerns. However, the following concern has not been appropriately addressed:

4. Also on the mechanism theme, how the SARS-CoV-2 Spike protein (S) induces lung fibrosis is not really addressed in the study. What regions of the protein are important and why?

While the first part of this important point was addressed, the second part: “What regions of the protein are important and why?” has not been addressed experimentally. Even very rough stalk vs. head vs. cytoplasmic/transmembrane regions could be addressed with not too much effort, for example by adding soluble S vs S stalk protein vs S head protein. Such an experiment would also allow for the clotting phenotype observed to be most definitive.

Response: We thank the reviewer for recognizing our study as “definitely an important and timely topic of research”. As to the remaining question raised by the reviewer regarding to “What regions of the protein are important and why?”, we provided new results in the last revision to show that in addition to SARS-CoV-2, VSV infected NHBE cells also supported fibrin clot formation, albeit less profound than that produced by SARS-CoV-2 infection. This suggests that the infection-induced fibrin clot formation depends on the viral infection of NHBE cells but may not be unique to SARS-CoV-2, which enters via ACE2 binding. This is also consistent with the failure of fibrin clot formation when using env-null virus to infect NHBE cells, which is at least 100x less infectious than the spike-containing virus. Nevertheless, it is possible that soluble spike, RBD or other regions of the viral spike protein binding to NHBE cells may signal NHBE cells leading to fibrin clot formations. To address the reviewer’s comment, we incubated NHBE cells with soluble spike trimer, S1 subunit, or RBD and assayed the fibrin clot formation after incubation. Concurrently, we performed pSARS-2 (omicron strain) or mock infection of NHBE cells as controls. As expected, the infected but not uninfected NHBE cells induced fibrin clot formation (Supplemental Figure 2I). However, no fibrin clot formation was detected in the soluble spike, S1 subunit or RBD treated cells. This result suggests soluble spike binding to NHBE cells (including ACE2 receptor) is not sufficient to induce fibrin clot formation, and the NHBE cell-induced fibrin clot formation requires viral infection. Regions of the spike protein may be important as they influence the viral entry.

We included the new data in Supplemental Figure 2I and revised manuscript accordingly. The following paragraph is added to the results section:

“To further address if SARS-CoV-2 spike alone or the RBD binding to ACE2 on NHBE cells are sufficient to induce fibrin clot formation, we treated NHBE cells with soluble Wuhan spike trimer, S1 subunit or the ACE2 binding RBD domain and performed fibrin clotting assay after the treatment. While the omicron pSARS-2 infected NHBE cells induced fibrin clot formation,

none of the soluble protein treated NHBE cells supported fibrin clotting (Supplemental Figure 2I), suggesting that spike binding to NHBE cells is insufficient and viral infections are required for NHBE cells to induce fibrin clotting. The fibrin clot formation by infected NHBE cells may not be unique to SARS-CoV-2, however, as VSV-pseudotyped virus infection of NHBE cells also induced fibrin clotting (Supplemental Figure 2G).”

Comments from Reviewer #2 (Remarks to the Author):

The authors have provided comprehensive and adequate responses to my queries. There are no additional comments

Response: We appreciate the effort by this reviewer and are glad to see no further comments from the reviewer.

Comments from Reviewer #3 (Remarks to the Author):

The authors were partially responsive to the questions and concerns raised during the initial review of the manuscript. Important additional data were provided using the NHBE cell culture system. Most notably the authors provide data suggesting TF does not contribute to fibrin formation in this system. There is now also data using human prothrombin that is compatible with the proposed mechanism. It is clear that there is prothrombin, fibrinogen, and other coagulation proteins in BALF fluid from infected individuals and that this material has the capacity to form fibrin. However, a major weakness of the manuscript remains. Namely, there is no data to suggest that the proposed mechanism of TTSP mediated activation of prothrombin occurs in the airway in vivo, as proposed in Supplement Figure 7. The authors could not document the presence of TTSP proteins in patient BALF (infected or non-infected). Moreover, it appears that there is not an animal model system amenable to testing the hypothesis. Finally, the last paragraph of the discussion accurately notes that heparins, which primarily target factor Xa but also thrombin, have not been beneficial in mitigating COVID associated acute lung damage. Here the authors suggest that this lack of efficacy is due to TTSP proteins bypassing factor X to activate thrombin. This entire argument is predicated on the concept that fibrin deposition in the alveolar space exacerbates acute lung injury following SARS-CoV2 infection. Whereas this may well be the case, this reviewer is not aware of any data definitively demonstrating this to be true. The fact that animal models are not amenable to testing the author’s hypothesis put forth in the present study also precludes direct studies demonstrating a causative role for fibrin in driving DAD. At minimum, each of these weaknesses should be better addressed in the Discussion section of the manuscript.

Response: We would like to clarify the issue whether TTSP can be found in our BALF samples. We previously did not find ST14 and TMPRSS11D from mass spectrometry analyses of COVID BALF. Since both are membrane proteins with glycosylation, that are known difficult for peptide analysis by mass spectrometry due to poor fragmentation and glycan heterogeneity. In addition, only the C-terminal shed fragments are expected in BALF. The use of full length protein sequence in the reference database would underestimate their peptide coverage. When we use a less stringent cutoff in mass spectrometry analysis, both ST14 and TMPRSS11D were found with multiple peptides mapping to their extracellular

regions in the COVID BALF samples. Thus, not finding TTSP in the previous analysis is likely due in part to a technical issue related to these proteins. The abundances of these TTSP appear much lower than those of fibrinogen-subunits (Table S1). We revised the discussion section to reflect this: “While the infected cell culture supernatants exhibited enhanced cleavage of prothrombin peptide and the infected NHBE cells shed soluble ST14 (Figure 5B, 6C), trace amount of the extracellular fragment of ST14 and TMPRSS11D could be found in the COVID BALF samples (Table S1).” We also revised the method section to note the use of a less stringent criteria for finding TTSP.

While our data supports TTSP activation of prothrombin from their cleavage of the prothrombin peptide, we do appreciate the reviewer’s comment on the importance of using appropriate animal models for *in vivo* validations. We have revised the discussion section following the reviewer’s comments to both discuss the animal model issue and the importance of validate our *in vitro* mechanism in future using *in vivo* models. The revised discussion includes the following sentences:

“It is worth noting that while SARS-CoV-2 infected primary human lung epithelial cells induced fibrin clot formation in acute COVID lung fluid, it remains to be seen if this occurs *in vivo*. However, despite the presence of several SARS-CoV-2 susceptible animal models, including various ACE2-transgenic mice, Syrian hamster and rhesus macaque^{67, 68, 69}, they generally lack viral ARDS-like lung pathology observed in fatal human cases⁷⁰. They are valuable for vaccine and antiviral development, but the infected animals rarely develop DAD that is characteristic of ARDS. Our study here is primarily based on an *in vitro* primary lung cell model with the results validated using *ex vivo* COVID-patient BALF. The lack of a suitable animal model hampers further *in vivo* validation of the cellular mechanism linking fibrin deposition, hyaline membrane formation and DAD.

It would be interesting to see in a suitable animal model if nebulization *in vivo* indeed offers a better therapeutic outcome than intravenous injection in treating SARS-CoV-2 infected lungs.”

Reviewers' Comments:

Reviewer #1:

Remarks to the Author:

The authors have revised the manuscript according to my prior suggestions for revisions at this point.

Reviewer #3:

Remarks to the Author:

The authors have responded to the comments from my previous review. I have no new comments.

We thank the reviewers for their in-depth review of our manuscript and for their constructing comments. As a result of our revision, all three reviewers are satisfied with the current version of our manuscript.

REVIEWERS' COMMENTS

Reviewer #1 (Remarks to the Author):

The authors have revised the manuscript according to my prior suggestions for revisions at this point.

Reviewer #3 (Remarks to the Author):

The authors have responded to the comments from my previous review. I have no new comments.